



# Improving the Representation of Aggregation in a Two-moment Microphysical Scheme with Statistics of Multi-frequency Doppler Radar Observations

Markus Karrer[a], Axel Seifert[b], Davide Ori[a], and Stefan Kneifel[a]

[a]Institute for Geophysics and Meteorology, University of Cologne, Cologne, Germany
[b]Deutscher Wetterdienst, Offenbach, Germany

**Correspondence:** Markus Karrer (markus.karrer@uni-koeln.de)

**Abstract.** The simulation of aggregation of ice particles is critical for precipitation prediction, but still a major challenge. Its simulation requires assumptions about numerous parameters, many of which are either not well known or difficult to represent accurately in bulk microphysics schemes. However, knowing the sensitivity of aggregation to various simplified assumptions can help to identify critical parameters. By comparison with suitable observations, these critical parameters can
even be constrained. We investigate the sensitivity of the model variables, and the modeled multi-frequency and Doppler radar observables to different parameters in a two-moment microphysics scheme. Therefore, we revise hydrometeor parameters by using a recently published dataset of particle properties, modify the formulations of the aggregation process (which allows using an area-based differential sedimentation kernel) and update other ice microphysical parameters in the scheme such as the sticking efficiency $E_{\mathrm{stick}}$ and the shape of the size distribution. Overall, particle properties, definition of the aggregation
kernel, and size distribution width prove to be most important, while $E_{\mathrm{stick}}$ and the cloud ice habit have less influence. Finally, we run multi-week simulations with the most promising parameter combinations. The statistical comparison between real and synthetic observables shows a reduction in the velocity and snow particle size. With this study, we show a possible way to revise processes in microphysical schemes by using statistics of detailed cloud radar observations.

## 1   Introduction

Ice growth processes which lead to precipitable particles are essential to understand because more than 60% of the global precipitation reaching the surface is formed in the ice phase (Heymsfield et al., 2020). Besides depositional growth and riming, aggregation is one of the key growth mechanisms in ice clouds. Aggregation is found to be active in a large temperature range (Hobbs et al., 1974; Kajikawa and Heymsfield, 1989; Field, 2000). As revealed for example by radar observations, aggregation can cause a rapid increase of the particles size in favorable conditions, such as the dendritic growth zone or close to the melting

layer (Lamb and Verlinde, 2011). Unlike depositional growth, sublimation or riming, aggregation does not directly modify the ice and snow water content. However, its strong influence on particle shape, particle size distribution, and terminal velocity $v_t$ link aggregation to other processes, such as depositional growth, sublimation and riming that alter the mass flux considerably. Therefore, it is important to accurately represent aggregation in microphysical schemes.





A central component of the theoretical description of aggregation (see also Sect. 3.1) is the aggregation kernel. Therefore,
many challenges in accurately simulating aggregation can be discussed by considering the various components of this kernel.
The aggregation kernel is defined analogously to collision-coalescence of droplets in liquid clouds:

$$K(D_i, D_j) = \frac{\pi}{4}(D_i + D_j)^2 |v_t(D_i) - v_t(D_j)| E_{\mathrm{coll}}(D_i, D_j) E_{\mathrm{stick}}(T) \tag{1}$$

The aggregation kernel is proportional to the probability $K$ of two particles $i$ and $j$ to collide (Gillespie, 1975) and stick
together after collision. This probability increases with increasing size $D$ and relative $v_t$ of the particles, as well as the collision
$E_{\mathrm{coll}}$ and sticking efficiency $E_{\mathrm{stick}}$. Obviously, the size $D$ is well defined for spherical particles by their diameter, but this is
already much more complex for ice and snow particles which have a non-spherical shape. How large $v_t$ of ice and snow particles
is, also strongly depends on their size, shape and orientation (Böhm, 1992; Mitchell and Heymsfield, 2005; Heymsfield and
Westbrook, 2010). While for large aggregates $v_t$ does not increase with size (Lohmann et al., 2016) and is close to 1 m/s, for
smaller particles $v_t$ increases strongly with size. The size ranges where $v_t$ is most sensitive to the particle size (i.e. has the largest
slope) are highly shape-dependent (Barthazy and Schefold, 2006; Hashino and Tripoli, 2011; Karrer et al., 2020). Consequently,
the slope of the $v_t$-size relation is uncertain, but at the same time crucial for aggregation. Two remaining parameters, $E_{\mathrm{coll}}$ and
$E_{\mathrm{stick}}$, are also multiplicative in the kernel. $E_{\mathrm{coll}}$ describes the ratio between the actual collision cross-section and the geometric
cross-section. $E_{\mathrm{coll}}$ is smaller than one for most particle pairs because typically the smaller and slower particle gets deflected
due to hydrodynamics as the larger particle approaches. Compared to collisional processes involving cloud droplets, $E_{\mathrm{coll}}$ is
less important for aggregation because very small ice particles that could be strongly deflected are rare in temperature ranges
where aggregation dominates. Under some conditions, e.g. with active secondary ice production, $E_{\mathrm{coll}}$ should be considered.
However, $E_{\mathrm{coll}}$ is poorly constrained for ice-ice collisions (Wang, 2010). $E_{\mathrm{stick}}$ is the probability of two ice particles to stick
after the collision. Although laboratory (Hosler and Hallgren, 1960; Connolly et al., 2012; Phillips et al., 2014) and in-situ
(Mitchell, 1988; Kajikawa and Heymsfield, 1989) estimates, and multi-frequency radar retrievals (Barrett et al., 2019) exist,
the value of $E_{\mathrm{stick}}$ is very uncertain. However, there is widespread agreement in the literature on two main temperature ranges
in which $E_{\mathrm{stick}}$ is large: Around -15 °C, the mechanical interlock of dendrites increases $E_{\mathrm{stick}}$ compared to the surrounding
temperature regions (Pruppacher and Klett, 1998). In addition, sintering of ice particles due to an increasingly thick quasi-
liquid layer (Slater and Michaelides, 2019) on the ice surface cause a general increase of $E_{\mathrm{stick}}$ when temperature rises up to
0 °C. In addition to the aggregation kernel, the aggregation rate is also influenced by the particle size distribution. Simply put,
the particles that have a high probability of aggregation, given by the aggregation kernel, must be present in the cloud to have
efficient aggregation.

Bulk microphysics schemes can not simulate aggregation on an individual particle level, but require the calculation of bulk
aggregation rates. Analytic solutions for the bulk aggregation rates are in principle possible (Verlinde et al., 1990). However,
these solutions are computational expensive and require the usage of power law relationship for $v_t$ and size, which can not
represent the asymptotic behavior known from observations for large sizes. Approximations of the bulk aggregation rates





consider characteristic velocity differences (Wisner et al., 1972; Seifert and Beheng, 2006) and allow to use Atlas-type $v_t$-size relations, which consider the asymptotic behavior of $v_t$ at large sizes and non-spherical particle shapes (Seifert et al., 2014).

In general, we need to distinguish between three different aspects of uncertainty in aggregation simulations: 1) A general lack of understanding or quantification of parameters, such as the absolute values of $E_{\text{stick}}$. 2) Formulation of functional relationships, which can not adequately represent the whole relevant range (e.g. $v_t$-size relationship). 3) Simplifications that must be made to keep the computational cost affordable e.g. considering only bulk properties of the particle population. Because of these uncertainties, it is important to constrain the model by observations of aggregation in clouds.

In situ and remote sensing observations have provided valuable information on the general characteristics of aggregation and have allowed estimation of the relative importance of aggregation with respect to other processes. Already decades ago, observations reported that the largest aggregates are found around -15 °C, which is considered to be a consequence of mechanical interlocking of dendrites, and at temperatures a few degrees below 0 °C, which is related to the quasi-liquid layer (Lamb and Verlinde, 2011). Radar observations contain valuable information about the aggregation process, which also is the reason we rely on them in this study. The strong temperature dependence of aggregation observed in early studies could be confirmed by radar observations, especially in profiles of absolute and differential reflectivity (Kennedy and Rutledge, 2011; Andrić et al., 2013; Schrom and Kumjian, 2016; Moisseev et al., 2015). By considering, additionally, the mean Doppler velocity, the relative importance of aggregation and riming can be estimated (Mosimann, 1995; Mason et al., 2018; Kneifel et al., 2020). Furthermore, using radars of different frequencies allows to estimate mean particle size (Matrosov, 1998; Hogan et al., 2000; Liao et al., 2005; Szyrmer and Zawadzki, 2014; Kneifel et al., 2015) and therefore to better characterize at which temperatures aggregation is dominant.

Ori et al. (2020a, O20) evaluated ice particle growth in simulations of the Icosahedral Nonhydrostatic Model (Zängl et al., 2015, ICON) using the Seifert-Beheng two-moment microphysics scheme (Seifert and Beheng, 2006, SB06) by comparing it with measurements in observational space. To this end, O20 used the multi-month cloud radar dataset from Dias Neto et al. (2018, D18). This quality-controlled dataset is particularly suitable because it contains multi-frequency and Doppler measured and thus fingerprints of aggregation and sedimentation. While model-observation comparisons based on a single or few cases can be difficult to interpret due to the specific conditions (specific water vapor field, synoptic situation) of the case, the statistical comparison of O20 could reveal model-inherent mean biases. O20 found an overall correct temperature dependency of aggregation but also revealed an overestimation of the snow size and $v_t$ at temperatures above -7 °C. O20 suggested that inaccurate $E_{\text{stick}}$ and $v_t$-size parameterization might cause this overestimation. However, direct attribution of the observed biases (e.g. too large snow) to a specific component of the aggregation process (e.g. $E_{\text{stick}}$), requires simultaneous investigation of the influence of all parameters important for the aggregation process in a suitable modeling setup.

Microphysics schemes are usually tuned to improve the prediction of key variables, such as precipitation, the energy balance at the top of the atmosphere or the near-surface temperature (Schmidt et al., 2017; Morrison et al., 2020). Only a small subset of variables (e.g., $v_t$ of cloud ice) are varied during the tuning process and tuning might be ad hoc rather than evidence-based. As the models simulate complex interacting processes, several parameter combinations can improve the predicting skill of modeled variables such as precipitation. Therefore, it is likely that tuning introduces compensating errors. For example, if two parameters



are not accurately implemented, adjusting one of them might improve the model performance even when the adjustment leads away from the true value of the parameter. Detailed remote sensing observations can be used to adjust parameters and make improvements on the process level rather than improving the performance of the entire modeling system. However, because remote sensing observations are sensitive to a limited number of parameters and within a limited range of variability, there is a risk that model parameters may be adjusted to match observations well but still be inaccurate in regimes where these observations have low sensitivity. To reduce this risk, new methods for model improvement and development have been proposed which parameter selection is still based on physical constraints, namely theory and laboratory measurements, but can be optimized by Bayesian inference of observations (Morrison et al., 2020). The advantage of this approach is that uncertainty of both laboratory measurements and remote sensing observations can be considered and new knowledge about parameters can be continuously incorporated. In addition, novel cloud radar techniques, e.g., multi-frequency Doppler observations, enable the identification of key growth mechanisms (Kneifel et al., 2015; Kalesse et al., 2016; Pfitzenmaier et al., 2018; Barrett et al., 2019). Barrett et al. (2019) identified a temperature range where aggregation rapidly increases particle size and estimated $E_{\mathrm{stick}}$ from a retrieval using multi-frequency Doppler spectra. Identifying a dominant growth mechanism allows focusing on a single process, which simplifies the inverse problem by reducing the number of parameters and observables to be considered simultaneously.

In this study, we constrain the parameters that influence aggregation by confronting idealized and realistic simulations with the multi-frequency Doppler radar observations from D18. Methods used are described in Sect. 2. We revise all main parameters and functional relationships regarding the aggregation formulation in SB06 by incorporating recently published parameters and revising the bulk aggregation equations. We describe these parameters and formulations in Sect. 3.1 and compare them with the choices in the default SB06 scheme. In Sect. 3.1.5 the selection of the snow particle properties, which is a critical component of both aggregation and radar simulations, is described. The sensitivity of the aggregation and associated radar variables to individual parameters of the revised formulation is evaluated with an ensemble of 1D model simulations (Sect. 3.2). The optimal combination of these simulations is chosen and tested in sensitivity studies in ICON-LEM simulations (Sect. 3.3). Finally, we perform ICON-LEM simulations of several weeks, which we evaluate against the default simulations from O20 and the observations from D18 (Sect. 3.4). This approach allows testing many different parameters against observed statistics of several weeks in a numerically efficient way. Sect. 4 summarizes the approach and concludes on the following questions: How can we investigate the sensitivity of aggregation to the components of its parameterization? How can we improve the representation of aggregation in a two-moment microphysical scheme? Which microphysical parameters influence the simulation of aggregation the most?

## 2  Methods

The Icosahedral Nonhydrostatic Model (ICON; Zängl et al., 2015) has numerous applications due to its different configurations. ICON-NWP (ICON-numerical weather prediction) is used by Deutscher Wetterdienst (DWD) for operational weather forecast in a global and recently also in all regional setups. ICON's large-eddy mode is called ICON-LEM (Dipankar et al., 2015; Heinze





et al., 2017). We use the SB06 two-moment microphysics scheme instead of the single-moment scheme currently used in
operational weather forecasting, as do most studies that perform simulations with ICON-LEM. Since simulations with ICON-
LEM are relatively computationally expensive, we also use a simple 1D model to efficiently test different parameterizations
and their influence on the simulation.

Since we want to further investigate the causes and reduce the discrepancies between modeled and simulated observables, we
use the same simulation setup of ICON-LEM as in O20. We only briefly describe the setup here, since an extensive description
can be found in O20. The modifications we make to the SB06 microphysics scheme are described in detail in Sect. 3.1.

## 2.1 "Snowshaft" Model

Simple 1D models have been used to assess the influence of several parameters or processes on microphysical or observed quantities (e.g. precipitation rates, polarimetric variables) and to test new parameterizations (Seifert, 2008; Kumjian and Ryzhkov, 2010; Milbrandt and Morrison, 2016; Paukert et al., 2019). These models are much simpler than full 3D models (like ICON-
LEM) and are therefore also referred to as rainshaft models. Because we apply such a simple model to ice microphysics we use the term "snowshaft" model. In these simple models, the atmospheric variables (e.g. temperature gradient, relative humidity) are predefined and feedback mechanisms from microphysics to thermodynamic and thus dynamic variables are neglected. These simplifications allow the analysis of selected processes and their sensitivity to a range of parameters without having to consider the full range of complexity. Another advantage of the "snowshaft" model is the low computational effort which
allows testing a large number of parameter combinations and process formulations.

The "snowshaft" model has 250 layers and the temperature spans the range from 0 to -40 °C, which covers the most relevant range for precipitating ice clouds. The temperature profile is linear with a gradient of $0.0062$ Km$^{-1}$. Consequently, the top of the model is at 6450 m. The relative humidity with reference to ice ($RH_i$) is constant for h>3000m and increases linearly until it reaches $RH$=100% ($RH$ is the relative humidity with reference to water (Fig. A5)). The thermodynamic variables are
constant in time and there is no air motion. These simplifications can be justified by the nearly stationary nature of many clouds and the low vertical velocity seen in the dataset of D18.

At the top of the model, a gamma distribution (following the size distribution parameter as described in Table 3) is initialized for cloud ice and snow. Together with the size distribution parameter, the mass density $Q$ and the number density $N$ completely define the size distribution at the model top. Below the model top, the size distribution evolves through the following micro-
physical processes: Sedimentation, depositional growth, and aggregation. These processes are considered dominant below the cloud top (where nucleation is especially important) and above temperatures near the melt layer, where riming rates increase sharply (Kneifel and Moisseev, 2020). The values of $RH_i$, $Q$ and $N$ are chosen in Sect. 3.2.1 to match profiles of observables with substantial precipitation.

## 2.2 ICON-LEM Setup

In our simulations, we use a small domain setup of ICON-LEM. This setup has been shown to be both computationally efficient and to represent clouds well in various conditions (Marke et al., 2018; Schemann and Ebell, 2020; Schemann et al.,



2020). The domain is circular with a radius of 110km and the observational site Jülich Observatory for Cloud Evolution Core Facility (JOYCE-CF; Löhnert et al., 2015) is in the center. At JOYCE the TRIple-frequency and Polarimetric radar Experiment for improving process observation of winter precipitation (Tripex, D18) took place, which we use in the model-observation

comparison. The horizontal resolution of the simulations is ca. 400 m, the vertical resolution ranges from 20m at the surface to 380m at the model top. With a total of 150 vertical layers, the atmosphere is simulated up to a height of 21 km. Initial and lateral boundary conditions are taken from the ECMWF Integrated Forecasting System (IFS). Initialization is carried out each day at 00:00 UTC. IFS data is incorporated as forcing on the lateral boundary every hour.

### 2.3 SB06 Scheme

The SB06 scheme is used in the "snowshaft" simulations (Sect. 3.2) and as the microphysics scheme in the ICON-LEM simulations (Sect. 3.3 and 3.4). The SB06 scheme is a two-moment scheme, that simulates the evolution of the number density ($N = M^{(0)}$) and mass mixing ratio ($Q = M^{(1)} \cdot \rho_{air}$). $\rho_{air}$ is the air density and $M^n$ (Eq. (2)) are the moments of the mass distribution (Eq. (5)):

$$M^{(n)} = \int\limits_{0}^{\infty} m^n f(m) dm \tag{2}$$

The scheme simulates six different hydrometeor classes (cloud water, cloud ice, rain, snow, graupel, and hail). The conversion from one to another class is in general associated with a specific microphysical process. For example, if cloud ice forms aggregates, $Q$ and $N$ of cloud ice are converted to snow (Sect. 3.1). Therefore, it is consistent to assume properties of monomers for cloud ice and properties of aggregates for snow. The predefined particle properties of the default setting of the scheme are listed in Table 2 for each hydrometeor, along with the properties of the cloud ice and snow class alternatives

proposed in Sect. 3.1.

In the SB06 scheme, aggregation rates are the product of collision rates and $E_{\text{stick}}$, because $E_{\text{coll}}$ is assumed to be one. In the scheme, the variance approximation (SB06), based on the work of Wisner et al. (1972), provides a computational feasible analytical solution of bulk collision rates. The variance approximation of Seifert and Beheng (2006) avoids the usage of pre-calculated look-up tables (Seifert et al., 2014) and unlike Wisner et al. (1972), is able to estimate collision rates of

self-collection, i.e., aggregation within a particle class. The default SB06 scheme assumes power-law relations for the $v_t$-size relation in the calculation of the collision rates. The extension of the variance approximation of Seifert et al. (2014), which allows using Atlas-type $v_t$-size relations (Sect. 3.1.3) is applied in the SB06 scheme for the first time in this study.

Details of the components of the aggregation process considered in the SB06 scheme can be found in Sect. 3.1 and Appendix A.





### 2.4 Passive and Active Microwave radiative TRAnsfer tool (PAMTRA)

The Passive and Active Microwave radiative TRAnsfer tool (PAMTRA; Mech et al., 2020) is used to simulate synthetic radar observations. Microphysical properties are represented consistently in the SB06 scheme and PAMTRA (Table 2).

Throughout the study, we adopt the same scattering assumptions for each of the hydrometeor classes in the SB06 default scheme ("SB cloud ice", "SB snow", "cloud droplet", "rain", "graupel" and "hail" in Table 2). As in O20, we apply the Self-Similar Rayleigh-Gans approximation (SSRGA; Hogan and Westbrook, 2014; Hogan et al., 2017) and coefficients derived from 3D models of aggregates of plates for cloud ice and aggregates of needles for snow. In O20, the coefficients used for the snow class were slightly adjusted to closely match the observed triple-frequency signature. The SSRGA parameters of aggregates of plates are also used for the new cloud ice categories ("column" and "needle" in Table 2). For Mix2, SSRGA parameters derived from the same 3D models used for the determination of particle properties (Karrer et al., 2020, K20) are available (Ori et al., 2020b). Since we find little influence of SSRGA parameters in Sect. 3.1.5, we use the adjusted SSRGA properties of the aggregates of needles from O20 for theMix2 aggregates throughout the study to be consistent with 020, although using the SSRGA parameters derived from the same 3D aggregate models would be most physically consistent.

### 2.5 Multi-frequency radar approach

Like O20, we use multi-frequency observations to derive information about the aggregation process. Multi-frequency observations are useful to distinguish the size of particles, since the ratio of wavelength and particle size along with the particle density, are the main factors that determine their scattering properties. The scattering of particles much smaller than the wavelength can be well approximated by the Rayleigh approximation. For larger particles, however, the interference of waves scattered from different parts of the particles must be considered (Kneifel et al., 2020), which leads to differential scattering among the various frequencies.

The ratio between the reflectivities of two radars with operating wavelengths $\lambda_1$ and $\lambda_2$

$$DWR_{\lambda_1,\lambda_2} = \frac{Ze(\lambda_1)}{Ze(\lambda_2)} = \frac{\lambda_1^4 \int \sigma_b(m,\lambda_1)f(m)dm}{\lambda_2^4 \int \sigma_b(m,\lambda_2)f(m)dm} \tag{3}$$

quantifies the amount of differential scattering. DWR is called the dual-wavelength ratio, Ze is the reflectivity, and $\sigma_b$ is the backscattering cross-section. As can be seen from Eq. (3), DWR does not depend on the absolute concentration, but on the mean size and shape of the distribution $f(m)$ (Sect. 3.1.1). D18 calibrated the observed Ze's using disdrometer measurements and the DWR's by assuming negligible differential scattering at the cloud top. The latter calibration removes also the path-integrated attenuation so that the differential scattering due to non-Raleigh backscattering can be considered as the main cause of the DWRs. More specifically, the shorter wavelength radar experiences non-Rayleigh scattering for a larger fraction of the particle population. Therefore, the DWR correlates with the mean mass of the distribution. Mason et al. (2019) and others have shown that not only the mean mass, but also the shape of the distribution, the particle density, and the internal structure of the particles (through $\sigma_b$) can substantially affect the DWRs. Given the radars available in D18, we investigate the sensitivity of aggregation by analyzing $DWR_{X,Ka}$ and $DWR_{Ka,W}$. The subscripts W, Ka, and X denote the radar bands and, more





specifically, the wavelengths of 3.3, 8.6, and 31 mm. Each combination of wavelengths is sensitive to a different range of particle sizes. For example, $DWR_{Ka,W}$ is most sensitive to mean particle sizes of unrimed cloud ice and snow between 0.5 and 3 mm and $DWR_{X,Ka}$ between 1.5 and 15 mm (O20). Outside this sensitivity range, DWRs are zero (small mean size) or asymptotically approach (saturate) a DWR value (large mean sizes) that depends on the scattering properties of the particles present. More detailed information on the approach and its sensitivities can be found in O20.

Moreover, D18 reported that strong riming is rare in their dataset, so aggregation is the main contributor to particle growth and thus the increasing DWRs from cloud top to cloud bottom.

## 3 Results and Discussion

### 3.1 Ice Microphysical Parameters Influencing Aggregation

To interpret the following sensitivity experiments, we describe which parameters need to be considered in the simulation of aggregation in a bulk scheme, which parameters and process formulations are currently used in the SB06 scheme and how the assumptions could be updated with recently published parameterizations.

The stochastic collection equation (SCE) describes how the particle distribution ($PSD_m$) changes with time under the action of aggregation (Khain et al., 2015):

$$\frac{\partial f(m_i)}{\partial t} = \int_0^{m_i/2} f(m_j)f(m_i-m_j)K(m_i-m_j,m_j)dm_j - \int_0^{\infty} f(m_i)f(m_j)K(m_i,m_j)dm_j \qquad (4)$$

Here f(m) is the particle distribution as function of mass and K is the aggregation kernel described in Sect. 3.1.2. The first term of Eq. (4) describes the gain of particles of mass $m_i$ by aggregation of particles with masses $m_j$ and $m_i-m_j$. The second term considers the loss of particles of mass $m_i$ by aggregation with particles of mass $m_j$ (illustrated in Fig. 1 a and b). In general, PSDs can not be perfectly described by simple functional relationships (e.g. gamma distribution) but can have complex shapes (Fig. 1(a)). Thus, explicit prediction of the evolution of PSDs must take into account the full SCE.

Bulk schemes, however, can only account for the evolution of the PSD in a simplified form. The tendencies of the moments in the SB06 scheme (mass density: $\partial Q/\partial t$, number density: $\partial N/\partial t$) can be calculated by considering only the loss term. The reason for this can be further explained with Fig. 1(c-h), where the collision events are separated among the ice (monomers) and snow (aggregates) classes. In fact, because of the mass conservation, the total mass of particles gained (integral of the first term) has to match the total mass of particles lost (integral of the second term). Since it is assumed that within one timestep a particle can participate only in one collision event, only one snow particle results from the collision of two ice particles (number of arrows in Fig. 1(c and d)). The same applies for the ice-snow and snow-snow collisions but here there is no conversion of $N$ from one to another category but only a loss of $N_i$ or $N_s$. Thus, it is sufficient to calculate only one collision rate for each of the three considered collision scenarios (ice-ice, ice-snow, snow-snow) and moments ($N$ and $Q$).





**Figure 1.** Illustration of the SCE (Eq. (4)) for an explicitly resolved PSD$_m$ ((a) and (b)) and when applied to the cloud ice and snow classes of the SB06 scheme ((c) to (h)). The left column depicts the loss term (second term in Eq. (4)), the middle column the gain term (first term in Eq. (4)). The right column shows the sign and connection of the tendency of the bulk moments. Arrows indicate whether the number concentration is rising or falling at the specified mass. Red lines indicate the ice distribution, blue the snow distribution. The arrows are red if ice particles are collected and blue if snow particles are collected or are created as a result of the collision.



### 3.1.1 Size Distribution

In most bulk schemes, the PSD is described by the generalized gamma distribution or simplifications thereof. With the mass $m$ as a primary variable, the generalized gamma distribution can be written as:

$$f_m(m) \quad = \quad N_{0,m}\, m^{\nu_m}\, \exp(-\lambda_m m^{\mu_m}) \tag{5}$$

For some applications using the mass-equivalent diameter

$$D_{eq} = \left( \frac{6m}{\pi \rho_w} \right)^{1/3} \tag{6}$$

as a primary variable and the ordinary gamma-distribution is more convenient:

$$f(D_{eq}) \quad = \quad N_{0,eq}\, D_{eq}^{\mu_{eq}}\, \exp(-\lambda_{eq} D_{eq}), \tag{7}$$

where $D_{eq}$ is the mass-equivalent diameter. One such application is the use of the Atlas-type $v_t$-size relationship in the calcula-

tion of collision rates in Appendix A. Size distributions derived from in situ observations are usually presented as a function of the maximum dimension $D_{max}$, which is often derived by circumscribing a sphere or spheroid to the projected particle image:

$$f(D_{max}) \quad = \quad N_{0,max}\, D_{max}^{\mu_{max}}\, \exp(-\lambda_{max} D_{max}) \tag{8}$$

In general, a distribution described by Eq. (5) can not be expressed by Eq. (7) or (8). Only when $\mu_m = 1/3$ can Eq. (5) be expressed by Eq. (7). To allow conversion of Eq. (5) to Eq. (8), $\mu_m$ must be set to $b_m^{-1}$ (exponent in the $m$-$D_{max}$ relation;

Eq. (12)). As we calculate the collision rates of particles following an Atlas-type $v_t$-size relation (Appendix A), we need to set $\mu_m = 1/3$. Since $b_m \neq 3$ for cloud ice and snow, $\mu_{max}$ in Eq. (8) can only be approximated.

   The PSD's shape can vary strongly, e.g., for non-stationary events (Seifert, 2008). Furthermore, $\nu_m$, or equivalent parameters in distributions that use a different primary variable, is often described as a function of other parameters (e.g. the mean size; Heymsfield, 2003). Nevertheless, in the current version of the SB06 scheme, we must choose a single value of $\nu_m$ in each

simulation. Therefore, we test two different values of $\nu_m$ in the simulations and later (Sect. 3.2) select the one with which the simulations can reproduce the observations the best. SB06 standard configuration ($\nu_m = 0$) corresponds to $\mu_{eq} = 2.0$ and $\mu_{max} = -0.11$. If we use $\nu_m = 2$ instead of $\nu_m = 0$, we obtain a narrower distribution with much fewer particles at small masses and a peak near the mean mass. Considering Heymsfield (2003), $\nu_m = 0$ is representative of a mean mass diameter $D_{mean}$ of about 0.5 mm and $\nu_m = 2$ is representative for $D_{mean}$ of about 0.2 mm. Many studies have shown that the size distribution parameters

are correlated (e.g., Field et al., 2005; McFarquhar et al., 2015), further complicating the selection of $\nu_m$. Moreover, PSDs can exhibit bimodalities e.g. due to secondary ice generation (Korolev and Leisner, 2020), which can be accounted for by the two classes of cloud ice and snow in the SB06 scheme.

   The PSD width affects the aggregation rates and the radar variables. The narrower the distribution is, the lower are the aggregation rates. This is obvious from the bulk collision rates in Appendix A and can be explained by the small $v_t$ difference

of similarly sized particles (Sect. 3.1.2). The PSD width affects also the radar observables. The reflectivity, in Rayleigh-regime, is proportional to the 2nd moment of the PSD. A narrower distribution reduces the number of large particles (above $10^{-7}$kg





**Table 1.** Size distribution parameter for $\mu_m=1/3$, the mass-size relationship of the Mix2 particles (Table 2), q=$2\cdot10^{-4}kg\,m^{-3}$ and N=$10^4 m^4$ (same as Fig. 2). $Ze_{Ka}$, $MDV_{Ka}$ and $DWR_{X,Ka}$ are calculated using the self-similar-rayleigh-gans approximation (SSRGA) and the SSRGA parameters of Mix2 as provided by Ori et al. (2020b). $\mu_{max}$ is estimated by the zeroth, third and sixth moment of the distribution.

| $\nu_m$ | $\mu_{eq}$ | $\mu_{max}$ | $Ze_{Ka}$ [dBz] | $MDV_{Ka}$ [m/s] | $DWR_{X,Ka}$ [dB] | $DWR_{Ka,W}$ [dB] |
|---|---|---|---|---|---|---|
| 0.0 | 2.0 | -0.11 | 12.11 | 0.91 | 1.21 | 3.89 |
| 2.0 | 8.0 | 2.19 | 9.83 | 0.83 | 0.02 | 1.12 |

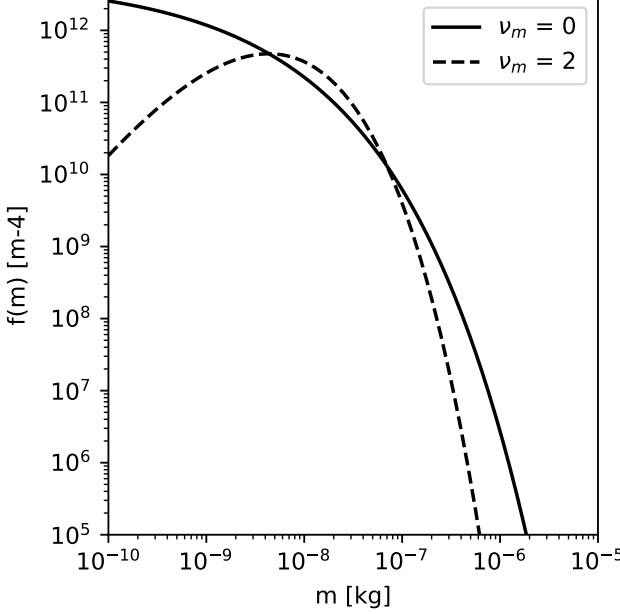

**Figure 2.** Particle distribution as a function of mass ($PSD_m$) with a mass concentration of $q = 2 \cdot 10^{-4}$ kgm$^{-3}$ and a number concentration of $qn = 10^4$ m$^{-3}$ illustrating PSDs with a different shape parameter $\mu_m$

in Fig. 2). Therefore, the reflectivity (Ze) and mean Doppler velocity (MDV) are slightly lower for a narrower distribution compared to a broader distribution with the same $Q$ and $N$. This effect is even stronger for DWRs, as the large particles contribute the most to the differential scattering signal (Table 1).

### 3.1.2 Collision Kernel

The D-kernel (Eq. (1)), defined analogously to the collisional coalescence of droplets in liquid clouds, is often used not only for particles that can be well approximated by spheres (e.g., cloud droplets, hail), but for all particles. However, the collision cross-section of non-spherical particles is smaller than the one of spheres with the same $D_{max}$ because of the presence of voids in their circumscribing sphere. This deviation was previously considered e.g. as a part of $E_{coll}$ (Keith and Saunders,



1989; Böhm, 1990) by using the equivalent circular radii $r_i = (A_i/\pi)^{0.5}$ as a characteristic length. Using the D-kernel with a constant $E_{\text{coll}}$ that does not dependent on particle size (as done e.g. in SB06) the D-kernel approximation can not account for the decrease in $A_r$ with increasing size (Fig. 3(d)). Therefore we test whether an alternative formulation of the collision kernel that takes the projected areas into account (A-kernel; Connolly et al., 2012) provides a better approximation:

$$K(D_i, D_j) = \left(A_i(D_i)^{0.5} + A_j(D_j)^{0.5}\right)^2 |v_i(D_i) - v_j(D_j)| E_{\text{stick}}(T) E_{\text{coll}}(D_i, D_j) \tag{9}$$

The A-kernel approximation has been used previously in the same or similar formulation (Kienast-Sjögren et al., 2013; Morrison and Milbrandt, 2015; Dunnavan, 2020). In these studies, the aggregation rates are calculated numerically, and in the case of the scheme proposed by Morrison and Milbrandt (2015), stored in look-up tables, that are used at the model run time. Seifert et al. (2014) argue that the use of look-up tables has disadvantages, like increasing complexity during preprocessing, additional memory access, and difficult reproducibility for follow-up studies. To avoid these disadvantages, the SB06 scheme

uses analytical solutions of the variance approximation introduced by Seifert and Beheng (2006). To use the A-kernel we have to generalize the collision rates. For brevity, we moved the lengthy derivations to Appendix A. To our knowledge, this is the first application of an A-kernel in a bulk microphysics scheme that uses an analytical formulation of the aggregation rates. How large the difference is between the D- and the A-kernel depends on the particle properties (e.g. area-size and $v_t$-size relation).

### 3.1.3  Particle Properties

Particle properties influence aggregation, because they are an essential part of the aggregation kernel. According to Eq. (1) and (9) collection is enhanced if the product of the difference in $v_t$ and the joint cross-section is large. Thus, a particle population will aggregate rapidly if the mean mass is relatively large and particles with largely different $v_t$ are present. The coefficients of area-size and $v_t$-size relations of the SB06 default scheme and the particle from K20 are included in Table 2.

While the particle properties of the SB06 default scheme particle classes are taken from in situ observations, K20 used

an aggregation model and hydrodynamic theory to simulate the particle properties. The advantages of this approach are that particle properties can be studied over a large size range, are physically consistent and can be studied in great detail. Particle property relations from in situ observations have a comparably small sample size. Thus, extrapolation to small and large sizes is unavoidable, because microphysics schemes need information about particle properties in a large size range. This extrapolation might lead to inaccuracies, such as the overestimation of $v_t$ at large sizes (K20). Since we take all snow particle properties ($m$-

size, $A$-size, $v_t$-size; Table 2) from the same aggregate type within the dataset, all properties are physically consistent. By comparing with in situ observations, K20 found that their mixed aggregates consisting of small columns and large dendrites (Mix2) can approximate mean aggregates properties well. Besides aggregates (including aggregates of columns and aggregates of dendrites, Sect. 3.1.5), K20 also summarized different monomer particle properties, e.g. the columns and needles shown in Fig. 3.

$v_t$ of the default cloud ice and snow class increase continuously with increasing size (Fig. 3) due to the used power-law relation:

$$v_t = a_{\text{vel}} m^{b_{\text{vel}}} \tag{10}$$





Due to this continuous increase, the self-collection rates of these hydrometeor classes stay relatively large at large sizes (Fig. A3 and A4). In contrast, the asymptotic approach to a limit of $v_t$ in the new relations leads to rapidly decreasing collision rates at large sizes. The asymptotic approach is evident from in situ observations and can be accounted for by using an Atlas-type $v_t$-size relation:

$$v_t = \alpha_v - \beta_v \exp(-\gamma_v D_{\mathrm{eq}}) \tag{11}$$

The relative $v_t$ of cloud ice and snow particles also play a role in ice-snow collection rates. In the SB06 default scheme, $v_t$ of cloud ice and snow differ greatly. However, K20 showed that $v_t$ of cloud ice and snow should have similar values. The difference between cloud ice and snow $v_t$ determines the location and magnitude of the minimum of the collection rates.

The projected area $A$ is derived differently in the D- and the A-kernel. In the D-kernel, the $m$-$D_{\mathrm{max}}$ relation

$$m = a_m D_{\mathrm{max}}^{b_m} \tag{12}$$

determines the relation between $A$ and size. Since $m$ is the primary variable in the SB06 scheme, it is most useful to consider the differences between the kernels and the particle classes as a function of $D_{\mathrm{eq}}$ (which is directly related to the mass).

$$A_{sphere} = \frac{\pi}{4} D_{\mathrm{max}}^2 = \frac{\pi}{4} \left( \frac{\pi \rho_w D_{\mathrm{eq}}^3}{6 a_m} \right)^{\frac{2}{b_m}} \tag{13}$$

Thus, the particles which have the lowest effective density

$$\rho_{eff} = \frac{6m}{\pi \rho_{ice} D_{\mathrm{max}}^3} \tag{14}$$

have the largest $A$ for a given $D_{\mathrm{eq}}$ (e.g. needles of K20 in Fig. 3(b)). The other particles have similar $A$. In the A-kernel, the actual projected area $A_{act}$, derived from the particle shape is relevant:

$$A_{act} = \gamma_A D_{\mathrm{eq}}^{\sigma_A} \tag{15}$$

The particle shapes and thus $A_{act}$ are not defined for the SB06 default classes because it does not require this property. The area ratio $A_r$ is commonly defined as the ratio of $A_{act}$ to the area of a sphere with diameter $D_{\mathrm{max}}$:

$$A_r = \frac{4 \gamma_A D_{\mathrm{eq}}^{\sigma_A}}{\pi D_{\mathrm{max}}^2} \tag{16}$$

At small sizes, $A_r$ is close to 1, indicating compact particles and small differences between the D- and the A-kernel (Fig. 3(d)). With increasing size, $A_r$ decreases down to 0.2 at $D_{\mathrm{eq}}$=5mm for the Mix2 class and lower for the cloud ice classes needle and column. $A_{act}$ is similar to observations of Mitchell (1996) as shown in K20.

However, the low values of $A_r$ of the cloud ice classes are less important because such large sizes of cloud ice are rarely reached in the model. The decrease in $A_r$ leads to a decrease of collision rates especially at large sizes, similar to the Atlas-type $v_t$-size relations. Thus, combining the new $v_t$-size relations with the A-kernel substantially decreases collision rates at large sizes.

While the properties of snow can be well validated against mean observed quantities (as done in K20 and in Sect. 3.1.5 of this study), selecting a single habit for cloud ice is a strong simplification, that is necessary for a simplified microphysics scheme.



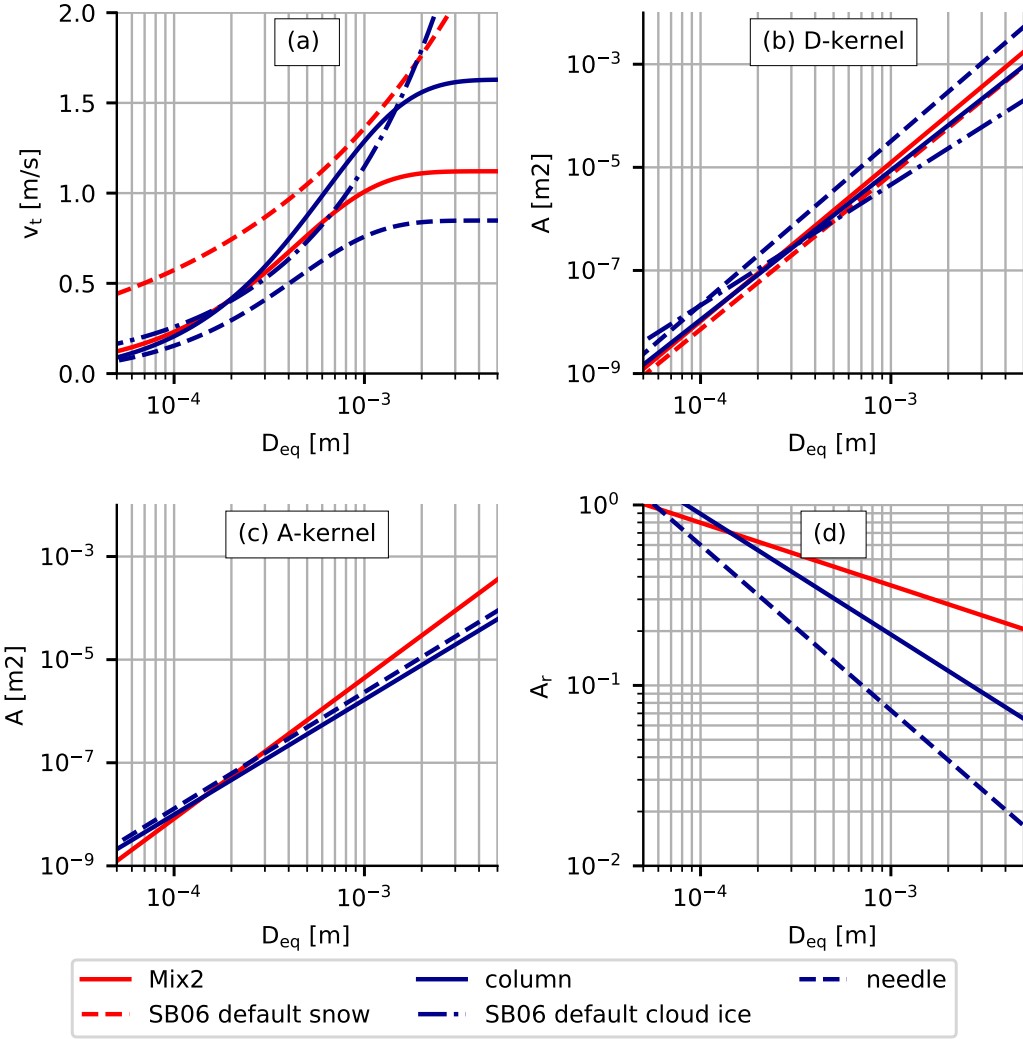

**Figure 3.** Particle properties from the default ("SB06 default cloud ice", "SB06 default snow") and modified version ("column", "needle", "Mix2") of the scheme. (a) Terminal velocity $v_t$, (b) Projected area $A$ of a circumscribing sphere (as assumed in D-kernel), (c) "Real" projected area $A$ considering the voids in the circumscribing sphere (as assumed in A-kernel) (d) Area ratio (Eq. (16)). The default scheme does not assume an A-D relation explicitly and therefore the "real" projected area and the area ratio are not given.

### 3.1.4 Sticking Efficiency

The parameters discussed so far determine how often collisions occur. The percentage of the colliding particles that stick together after a collision is defined by the sticking efficiency $E_{\text{stick}}$.

$E_{\text{stick}}$ is mostly only described as a function of the temperature (Mitchell, 1988; Connolly et al., 2012, M88,C12). To stick to each other, ice particles must form ice bonds (Lamb and Verlinde, 2011), which is highly unlikely for colliding solid-ice





**Table 2.** Parametrizations used in ICON-LEM, the "snowshaft" model and radar forward simulations of hydrometeor properties in PAMTRA. $D$ represent the particle maximum dimension and $D_{\text{eq}} = \left( \frac{6m}{\pi \rho_w} \right)^{1/3}$ the mass equivalent diameter. m is the particle mass and $\rho_w$ the density of water. The mass-size ($m$-D), terminal velocity $v_t$-size and projected area-size ($A$-D) relations are reported in their full mathematical form. For the SSRGA scattering model, the four parameters ($\kappa$, $\beta$, $\gamma$, $\zeta_0$) are given in parenthesis. SB indicates that the properties are exclusively used in the default setup. Cloud droplets, rain, graupel and hail (which are only relevant for the 3D- simulations) follow the same properties in all simulations. The aspect ratio is 1.0 for all classes except for the snow classes (SB snow, Mix2 and Mix2 (O20 scat)) where an aspect ratio of 0.6 is assumed. All variables are in SI units.

| Hydrometeor classes | m-D | A-D | v-D | Scattering |
|---|---|---|---|---|
| SB cloud ice | $1.588 D_{\max}^{1.56}$ | - | $30.6 D_{\max}^{0.55}$ | SSRGA(0.18,0.89,2.06,0.08) |
| Column | $0.046 D_{\max}^{2.07}$ | $8.21 D_{\text{eq}}^{2.23}$ | $1.63 - 1.67 e^{-1586 D_{\text{eq}}}$ | SSRGA(0.18,0.89,2.06,0.08) |
| Needle | $0.0047 D_{\max}^{1.89}$ | $13.97 D_{\text{eq}}^{2.26}$ | $1.41 - 1.43 e^{-1650 D_{\text{eq}}}$ | SSRGA(0.18,0.89,2.06,0.08) |
| SB snow | $0.038 D_{\max}^{2.0}$ | - | $5.51 D_{\max}^{0.25}$ | SSRGA(0.25,1.00,1.66,0.04) |
| Mix2 (O20 scat) | $0.017 D_{\max}^{1.95}$ | $685.93 D_{\text{eq}}^{2.73}$ | $1.12 - 1.19 e^{-2292 D_{\text{eq}}}$ | SSRGA(0.25,1.00,1.66,0.04) |
| Mix2 | $0.017 D_{\max}^{1.95}$ | $685.93 D_{\text{eq}}^{2.73}$ | $1.12 - 1.19 e^{-2292 D_{\text{eq}}}$ | SSRGA(0.22,0.60,1.81,0.11) |
| Aggregates of Columns | $0.074 D_{\max}^{2.15}$ | $69.34 D_{\text{eq}}^{2.50}$ | $1.583 - 1.6 e^{-1419 D_{\text{eq}}}$ | SSRGA(0.23,1.45,2.05,0.02) |
| Aggregates of Dendrites | $0.027 D_{\max}^{2.22}$ | $367.91 D_{\text{eq}}^{2.53}$ | $0.88 - 0.895 e^{-1393 D_{\text{eq}}}$ | SSRGA(0.23,0.75,1.88,0.10) |
| cloud drop | $\frac{\pi}{6} \rho_w D_{\max}^3$ | - | $2.49 \cdot 10^7 D_{\max}^2$ | Mie |
| rain | $\frac{\pi}{6} \rho_w D_{\max}^3$ | - | $9.3 - 9.6 e^{-622.2 D_{\text{eq}}}$ | Mie |
| graupel | $500.86 D_{\max}^{3.18}$ | - | $406.7 D_{\max}^{0.85}$ | soft-sphere Mie |
| hail | $392.33 D_{\max}^{3.0}$ | - | $106.3 D_{\max}^{0.5}$ | soft-sphere Mie |

particles the temperature is well below the melting temperature and the particles only touch for a short time. There are two
main mechanisms that increase the likelihood of adhesion after a collision and explain the temperature dependence. The first mechanism is explained by the quasi-liquid layer (QLL) on the ice particle surface. The phenomenon of QLL has been studied since the mid-19th century (Slater and Michaelides, 2019). QLL thickens with increasing temperature and consists of weakly bound molecules on the particle surface (Slater and Michaelides, 2019). When two particles touch, the molecules form a solid bond at the point of contact. The second mechanism is the mechanical interlocking of relatively large particles with dendritic
features (Pruppacher and Klett, 1998). These dendritic features occur at temperatures between -17 °C and -12 °C.

    The SB06 default scheme uses the $E_{\text{stick}}$ parameterization of Cotton et al. (1982) for ice-ice collisions and Lin et al. (1983) for ice-snow and snow-snow collisions (Fig. 4). The exponential shape of both parameterizations can be justified by the approximately exponentially increasing QLL thickness. These relations, however, miss the maximum of $E_{\text{stick}}$, suggested by studies (M88,C12), that consider the mechanical interlocking mechanism.

We combine M88 and C12 to propose a new parametrization. For $T <$-20 °C we follow C12, then linearly approach the plateau proposed by M88 with $E_{\text{stick}}$=1 between -17 °C and -12 °C. As discussed in the introduction, there is ample evidence from both in situ and remote sensing observations that $E_{\text{stick}}$ is likely to be present at temperatures near -15 °C (where particles





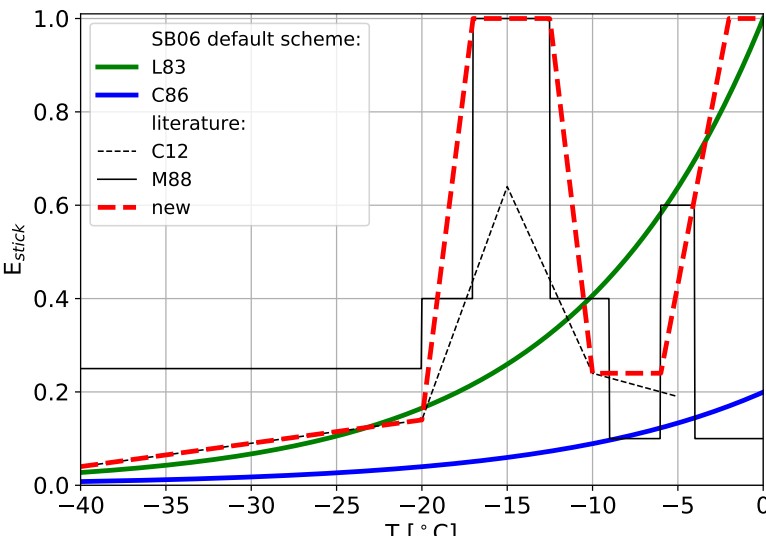

**Figure 4.** The sticking efficiency ($E_{\text{stick}}$) in SB06 scheme for collisions among ice particle (ice selfcollection) follows L83; for other collisions (ice-snow collection, snow selfcollection) it applies the C86 parameterization. Our new relation (red) combines the relations from M88 and C12 with a characteristic maximum around -15C and values quickly approaching unity for temperature larger than -5C.

with dendritic features are present) and near the melt boundary. At -10 °C the new parameterization again follows C12, but increases towards 1 at higher temperatures, where C12 does not provide an estimate of $E_{\text{stick}}$. One might prefer to follow C12

rather than M88, since C12 derived $E_{\text{stick}}$ directly from laboratory measurement and M88 provided only an ad-hoc parameterization. However, C12 analyzed only the initial stage of aggregation, where few monomers compose the aggregates. The interlocking mechanism might be more efficient for more complex aggregates compared to early aggregates as discussed in C12. Even considering only the initial stage of aggregation, the confidence interval of $E_{\text{stick}}$ at -15 °C ranges from 0.35 to 0.85 (C12).

**3.1.5 Selecting a Particle Type Representative for a Large Aggregate Ensemble**

After discussing the various components of the aggregation process formulation, we need to decide which aggregate type to use to best represent the physical particle properties (e.g. $v_t$) and scattering properties. In O20, the particle properties were defined by the assumptions in the standard SB06 scheme. The best-fitting aggregate model and associated SSRGA parameters were selected based on the best fit in the triple-frequency DWR space. In this section, we ask whether there is an aggregate

type in the database of K20 and Ori et al. (2020b) that reproduces well both the physical and scattering properties compared to the observations.

O20 already noted that the representation of MDV as a function of DWR resembles to some extent the underlying $v_t$-size relation. In contrast to the triple-frequency DWR-DWR, the MDV-DWR space is rather insensitive to the PSD width. Different





aggregate types composed of different monomer types generated and studied in K20 are used to simulate their corresponding

MDV-DWR signatures (Fig. 5). The underlying distribution shows the observed values from D18, which naturally contain
larger scatter and even negative DWR values, mainly due to imperfect radar volume matching (for a detailed discussion, see
D18). Fortunately, as shown in D18, the data set contains only very short and weak riming events. This scarcity of substantial
riming is important because the increased MDV due to riming would bias our comparison. Moderately or strongly rimed
particles would exceed 1.5m/s upon reaching a size that results in a nonzero $DWR_{Ka,W}$ (Mason et al., 2018). The MDV-

DWR space is also well suited to evaluate our aggregate choice, as it combines the two radar variables that showed the largest
discrepancies with the model simulations in O20.

O20 already recognized the overestimation of $v_t$ at large sizes, which is also evident in Fig. 5. For example, at $DWR_{X,Ka}$=5 dB
the observed MDV scatters around 1m/s, while the snow falls at $1.7~\mathrm{ms}^{-1}$ in the SB06 default scheme. From the aggregate
dataset of K20 the aggregates of dendrites fall the slowest and the aggregates of columns fall the fastest. A mixture composed

of small columns and large dendrites (Mix2), which fitted best to in-situ observations (K20), also matches the observations in
the MDV-DWR space well. Therefore, we utilize the Mix2 aggregate properties as an improved description for the snow class
in the following.

Interestingly, the use of the SSRGA coefficients of the aggregate type O20 does not lead to a strong change of the curves
in the MDV-DWR space. Although it would be most consistent to use the SSRGA coefficients of Mix2 directly, we will use

the scattering properties of O20 in the following analysis to allow a fair comparison of our new results with the discrepancies
found in O20.

### 3.2 Exploring Sensitivity to Microphysical Parameters in "Snowshaft" Model

The "snowshaft" model (Sect. 2.1) allows us to test the influence of the particle properties, the formulation of the collision
kernel, $E_{\text{stick}}$ and the size distribution on the aggregation rates with low computational effort and with reduced complexity.

In Sect. 3.1 we showed how these parameters affect aggregation. We do not only examine the influence of the parameters on
the predicted model variables but also on the radar observables. After carefully setting up the model, the comparison in radar
space enables us to directly contrast the statistics of the simulation and the observations, as given in O20 and D18. Since we
compare the statistics of the model and observations over a relatively long time range this analysis already attempts to select a
combination of parameters that can well reproduce the observational statistics. The optimal parameter combinations found in

the "snowshaft" simulations will then be applied in the 3D model to simulate a case study (Sect. 3.3) before we use it to rerun
simulations for the whole time period of the Tripex campaign (Sect. 3.4).

This comparison between model and observation benefits from the simultaneous consideration of multiple model parameters
and multiple observables. When looking at a single observable only, one might reduce a bias by an adjustment of a single
process or parameter, even though this might just compensate for an inaccurate choice in another parameter, introducing

compensating errors. As the number of independent observables increases, this problem is reduced as the inaccurate choice
of a parameter might be detectable in one of the remaining observables. In other words, the larger degree of freedom in the
observations helps to better constrain the parameters by comparison with the model when several observables are considered.





We focus our comparison on the DWRs (as a measure of particle size) and the MDV (as a measure of $v_t$). These two quantities constrain the strength of aggregation and the assumed $v_t$-size relationship and the statistical comparison in O20 also revealed the largest differences between observations and model in these variables.

### 3.2.1 Optimizing the "Snowshaft" Model and Selecting Microphysical Parameters for New Setup

O20 pointed out that the inconsistencies between observed and synthetic MDV and DWRs are especially evident for raining periods. As we attempt to remove these inconsistencies, the atmospheric variables and the hydrometeor contents at the top of the simulation are chosen so that the hydrometeor profiles in the "snowshaft" simulation roughly follow the profiles of the ICON-LEM simulations from O20 where $RR$ is larger than 1 mmh$^{-1}$; compare "default" and the histogram in Fig. 6. To match the profiles the $RH_i$ has to be set to 1% above about -18 °C and increasing values up to about 6% at about -7 °C. These values of $RH_i$, which are relatively high compared to those from the ICON-LEM simulations (Fig. A5), might be necessary because of the absence of nucleation and advection in the "snowshaft" simulations. Also, the values of $Q_i$, $N_i$, $Q_s$ and $N_s$ at the model top are chosen so that the hydrometeor profiles of the "CTRL" simulation (performed with the SB06 default setup) match well with those of the profiles of the ICON-LEM simulations of O20 with $RR > 1$mm (Fig. 6). After this optimization of the "snowshaft" model, the simulated profiles from ICON-LEM (O20 and Fig. 11 and 12) and the "snowshaft" model (Fig. 6) reveals that a simple initialization (nucleation) of the profiles at cloud top is sufficient at least for testing the sensitivities of aggregation to our set of parameters and various formulations.

After iterating over many parameter combinations, we found one particular setup (which we refer to as "colMix2_Akernel" or simply as "NEW") to match the observed profiles particularly well. In these iterations, we varied mostly the less-known components, e.g., the size distribution width, while parameters that we were already better able to constrain (Sect. 3.1.5), e.g., the $v_t$ size relation, were not varied. Our approach can hence be seen as a combination of a purely physically-based approach, incorporating current knowledge of parameters obtained e.g. through laboratory studies, and an empirical correction based on observations.

### 3.2.2 Sensitivity of Aggregation to Individual Ice Microphysical Parameters in "Snowshaft" Model

The hydrometeor profiles (Fig. 6) and radar observables (Fig. 7) of the "NEW" setup exhibit many interesting differences from the profiles of the "CTRL" run. In the following, we discuss where the differences originate from by looking at the different sensitivity runs. In each sensitivity run only one set of parameters is different from the "NEW" run (Table 3).

The cloud ice mixing ratio $Q_i$ and the cloud ice number density $N_i$ are lower in the "NEW" run than in the "CTRL" run for $T <$-10 °C (Fig. 6). At the same time, the snow mixing ratio $Q_s$ and number density $N_s$ are slightly larger in the "NEW" run at temperatures below -17 °C. These differences can be explained by the higher $E_{\text{stick}}$ at lower temperatures in the "NEW" setup (Fig. 4), which leads to more collisions among cloud ice particles and therefore more particles are converted from the cloud ice to the snow category. When using the $E_{\text{stick}}$ parameterization of Cotton et al. (1982) and Lin et al. (1983, "colMix2_Akernel_LinCot";), $Q_i$ and $N_i$ are larger at lower temperatures (and $Q_s$ and $N_s$ are smaller).



The smaller values of $E_{\text{stick}}$ in "colMix2_Akernel_LinCot" compared to "NEW" at lower temperatures (compare L83 and C86 with "new" in Fig. 4) lead to further differences. "colMix2_Akernel_LinCot" has a smaller mean mass $\bar{x}$, which is the mean mass of the sum of the cloud ice and snow class $((Q_i + Q_s)(N_i + N_s)^{-1})$, and correspondingly lower DWRs for $T <$ -7 °C (Fig. 7(c) and (d)). The smaller mean size also leads to slower falling particles (visible in $MDV$; Fig. 7(b)). For $T >$ -7 °C the strong increase of $E_{\text{stick}}$ in "colMix2_Akernel_LinCot" triggers a strong increase of $\bar{x}$ and $DWR_{X,Ka}$. A similar increase

in the mean and median of the investigated statistics of $DWR_{X,Ka}$ was already discussed in O20. As in O20 the strong increase is not visible in $DWR_{Ka,W}$, since this observable already reaches saturation for mass median diameters of about 3 mm (Sect. 2.5). The local maximum of the new $E_{\text{stick}}$ parameterization at temperatures from -17 °C to -12.5 °C leads in the "NEW" run to a rapid increase of the $\bar{x}$, DWRs, and MDVs in the same temperature range and therefore matches the observed profile of $DWR_{Ka,W}$ better than the "CTRL" run.

O20 speculated that the overestimation of the particle sizes at high temperatures and the mismatch in the profiles of the DWRs might be mainly due to the $E_{\text{stick}}$ parameterization and the $v_t$-size relation. However, Fig. 6 and 7 and the aggregation rates (Appendix A) reveal that the $v_t$-relation at smaller sizes and the aggregation kernel formulation also strongly affect the aggregation rates. Both $\bar{x}$ (Fig. 6 i) and DWR$_{X,Ka}$ are lower in "colMix2_Akernel_LinCot" than in the "CTRL" run. If $E_{\text{stick}}$ would be the dominating driver, these two simulations should be very similar. The differences in the $\bar{x}$ profiles of these two

simulations can only be explained by relevant influences of other parameters on the aggregation rates.

     The simulations with the D-kernel ("colMix2_Dkernel") exhibit a strong influence on aggregation. This is evident in the rapid decrease of $Q_i$ and $N_i$ and a rapid increase of $\bar{x}_i$, $\bar{x}_s$, and $\bar{x}$ caused by high aggregation rates (supported by Appendix A). From this simulation, it is evident that the use of the new particle properties (including the Atlas-type $v_t$-size relation) together with the D-kernel results in even larger particles than in the default run and thus DWRs are strongly overestimated (Fig. 7 d).

This overestimation can only be reduced by using the A-kernel.

     The vertical gradients of $Q$ result from mass uptake by depositional growth and divergence of $v_t$ (Fig. 6(h)). First, $Q$ increases from the cloud top to the cloud bottom due to depositional growth. Second, deposition growth and aggregation increase particle size and thus $v_t$ increases. If there were no mass uptake (no deposit growth) but only aggregation, $Q$ could only decrease because the product of $v_t$ and $Q$ would be conserved. The $v_t$-size relation plays an important role in these processes: on the

one hand, smaller $v_t$ for given particle size, e.g., as in "NEW" vs. "CTRL", means more time for mass uptake, leading to a faster increase in $Q$ per height. On the other hand, smaller $v_t$ could also lead to less ventilation and thus less mass uptake due to depositional growth. The $v_t$-size relationship, which defines the slope of $v_t$ with increasing size influences the divergence of $v_t$ with height and the aggregation rates (Sect. 3.1.3). These multiple effects also interact, which further complicates the interpretation of the profiles of $Q$. Nevertheless, we attempt to interpret the most obvious features of the profiles of $Q$.

At about -17 °C, MDV increases sharply in the "NEW" run (Fig. 7(b)), causing a decrease in $Q$ at these temperatures (Fig. 6(f)), while $Q$ increases continuously in the "CTRL" run. The differences in the profiles of $Q$ between the sensitivity runs are relatively large. These large differences are likely due to the different conversion rates of cloud ice to snow near and differently strong increasing $\bar{x}$ near the model top. For example, in "colMix2_Dkernel" the cloud ice converts rapidly to larger snow particles. As a result, particles near the model top fall faster and therefore have less time to grow by depositional growth


(the increase in $Q$ is weaker compared to the "NEW" run). "colMix2_Akernel_LinCot" shows a weaker increase in $\bar{x}$ for $T >$-15 °C compared to the "NEW" run (Fig. 6(i)). This weaker increase in $\bar{x}$ leads to a weaker increase in MDV (Fig. 7(b)) and thus to a stronger increase in $Q$ (Fig. 6(h)). The reflectivity $Ze_{Ka}$ is closely related to $Q$, so that "colMix2_Dkernel" ("colMix2_Akernel_LinCot") has the lowest (highest) reflectivity. However, the "CTRL" run has the highest $Ze_{Ka}$, although $Q$ is lower than in some sensitivity runs. The large $Ze_{Ka}$ here could be caused by the relatively dense snow particles assumed

in "CTRL" (Fig. 3). Overall, $Q$ and $Ze_{Ka}$ show relatively large sensitivity to the varied parameters in these "snowshaft" simulations. However, this observation must be interpreted with caution. The simulations assume a relatively large humidity in order to match the hydrometeor profiles and compensate for processes not considered (Sect. 3.2.1). This high humidity could lead to an overestimation of mass uptake due to depositional growth. Additionally, considering that supersaturation is not consumed by depositional growth but is held constant in our "snowshaft" simulations, one could hypothesize that $Q$ and

$Ze$ might be more similar among the sensitivity runs in the ICON-LEM simulations.

The new particle properties reduce the bias of the scheme regarding MDV to a large extent (Fig. 7b). While all simulations with the new particle properties are within the deciles of the observations, the standard run is already outside the deciles at -35 °C and is more than 0.5m/s larger than the median at some temperatures (e.g. at $T =$5 °C). The other parameters change the profile of the MDV to a much lesser extent. At temperatures from -18 °C to -12 °C, all simulations show an increase in MDV,

while all quantiles of the observed MDV decrease. This discrepancy could be due to the lack of habit prediction, underestimated or missing upwinds, or the lack of collisional fragmentation (Korolev and Leisner, 2020) in the model. At these temperatures dendritic growth occurs, which could lead to decreasing particle density and thus decreasing $v_t$ and/or updrafts as a result of strong latent heat release. Collisional fragmentation could furthermore lead to the formation of new, small particles with low $v_t$, which also reduces the MDV.

In addition to the particle properties, the width of the size distribution changes the MDV the most. The simulation with the wider size distribution ("colMix2broad_Akernel") has a larger MDV (Fig. 7b) than the "NEW" run, which is due to the increasing number of large particles at the larger end of the distribution (Sect. 3.1.1). These large particles contribute more to the MDV than the smaller particles; to calculate MDV, each particle must be weighted by reflectivity, which for Rayleigh scatterers scales approximately with mass to the power of two. The higher weight of the large participants also explains why

the DWRs in "colMix2broad_Akernel" are significantly higher compared to the "NEW" run, even though the mean size of the hydrometeors is relatively similar. This sensitivity illustrates, that the DWRs can only to some extent be used to infer $\bar{x}$ and the size distribution width has to be considered additionally.

Despite the various simplifications in the "snowshaft" model (no nucleation, no advection, constant humidity) the mean profile of the radar profiles from the ICON-LEM simulations of O20 could be well matched. This allowed us to investigate

the sensitivity of aggregation to the individual model components and to select a model setup that best matches the observed radar profiles. The particle properties of the snow, the aggregation kernel formulation, and $E_{stick}$ have a strong influence on the hydrometeor contents and the simulated radar observables. Interestingly, the choice of particle size distribution has little effect on the hydrometeor profiles but a large effect on the DWR values. The choice of cloud ice properties (needle or column) is less important than the choice of the other parameters in this cloud regime. However, the choice of cloud ice properties might





**Table 3.** Overview of parameters and settings varied in the microphysical sensitivity experiments. The sensitivity runs have the same settings as colMix2_Akernel unless otherwise noted. $K$ is the collision kernel, $D$ the maximum dimension and $A$ the particle's projected area, $\mu$ and $\nu$ are parameters in the generalized gamma function describing the mass distribution in the microphysics scheme (Eq. (5)).

| | Main runs | | Sensitivity runs (difference to colMix2Akernel) | | | |
|---|---|---|---|---|---|---|
| | SB06 default/ CTRL | colMix2_ Akernel/ NEW | needMix2_ Akernel | colMix2_ Dkernel | colMix2_ Akernel_LinCot | colMix2 broad_Akernel |
| Particle Properties (Fig. 3) | SB06 default cloud ice, SB06 default snow | Column Mix2 | Needle | | | |
| Collision kernel | D-kernel: $K\propto(D_i+D_j)^2$ | A-kernel: $K\propto(A_i^{0.5}+A_j^{0.5})^2$ | | D-kernel | | |
| Sticking efficiency (Fig. 4) | L83/C86 | Modification of M88 | | | L83/C86 | |
| Size distribution $N(m)=A\,m^\nu e^{-\lambda m^\mu}$ | $\nu=0$ (cloud ice & snow) | $\nu=2$ (cloud ice & snow) | | | | $\nu=2$ (cloud ice) $\nu=0$ (snow) |

be more important for clouds with smaller aggregation rates, e.g., cirrus. If we combine the A-kernel, the particle properties of Mix2 from K20, the newly proposed $E_{\text{stick}}$ parameterization and a relatively narrow size distribution the observed profiles of MDV and DWRs could be better matched. To test whether these sensitivities and improvements in "NEW" are also found persistently in more realistic simulations, we test in the next section whether these observations occur similarly in the ICON-LEM simulations.

**3.3 ICON-LEM Case Study Simulation Using the New Parametrizations**

In the "snowshaft" simulations (Sect. 3.2) we had to use several idealized assumptions. ICON-LEM (Sect. 2.2) contains additional processes (e.g. advection, nucleation, varying humidity field) and therefore simulates a more realistic representation of the atmosphere. In this section, we investigate the impact of the various parameters, studied in the sensitivity analysis, in a





more complex case study with a ICON-LEM simulation. Furthermore, the ICON-LEM simulations provide an opportunity to
extend the analysis to various conditions (e.g. non-stationary regime during the frontal passage, sublimation layers).

The case-study of interest was 3 January 2015, when a low-pressure area over the British Isles and an accompanying frontal system over western and central Europe determined the synoptic situation over the modeled domain. Shallow mixed-phase clouds are present in the morning and dissipate around noon (Fig. 8(a)). The passage of a warm front manifests itself at 10:00 UTC, first in high clouds and then in sinking cloud bases. These frontal clouds start to precipitate at 18:00 UTC. The selected
case is particularly interesting because it contains clouds in different regimes and precipitation of weak to moderate intensity.

The observed and simulated $Ze_{Ka}$ fields match relatively well for all simulations in terms of cloud structure and precipitation (Fig. 8). Both the shallow mixed-phase clouds and the frontal cloud are very well captured in terms of temporal and spatial structure.

$Ze_{Ka}$ exhibits strong differences between the observations and the simulation only in the rain and ice slightly above the
melting temperature in the period from 19:00 UTC to 23:00 UTC. The sharp decrease of the observed $Ze_{Ka}$ indicates strong sublimation. The presence of sublimation is also revealed by the model showing subsaturated air in this time range (Fig. A6). There are three main reasons that explain why the model does not accurately represent the sharp decrease in $Ze_{Ka}$ in this sublimation scenario. First, the humidity could be overestimated in the model, e.g., due to inaccurate forcing data. Second, particle sizes could be overestimated due to processes in microphysics that weaken the effect of sublimation. We can not
completely rule out the humidity mismatch, but we found a good agreement between the model and radiosonde data when available. Unfortunately, there was no radiosonde launched on the considered day. Thus, we are confident with the general ability of the model to accurately simulate the humidity field, but we can not rule out that inaccuracies in the simulated humidity field contribute to the bias in $Ze_{Ka}$. Lastly, also the parameterization of sublimation could be an error source. For example, the evolution of the PSD during sublimation is challenging to represent in a two-moment scheme (Seifert, 2008).
Since all of these reasons might be able to explain the mismatch in $Ze_{Ka}$, we should be cautious in assessing the validity of the assumptions of the individual model settings based on this sublimation feature. However, regardless of the accuracy of the model in predicting the humidity or simulating sublimation the following differences in $Ze_{Ka}$ of the model simulations underscore the importance of accurate prediction.

While the "NEW" (Fig. 8(c)) and most sensitivity runs show a slight decrease in $Ze_{Ka}$ due to sublimation in the time period
where the air is subsaturated, the sublimation is barely seen in $Ze_{Ka}$ of some other simulations (e.g "CTRL", "colMix2_Dkernel"; Fig. 8(b) and (g)). The differences between the simulations are caused by the differences in the particle size indicated by $DWR_{X,Ka}$ (Fig. 9). Similar to the "snowshaft" simulations, $DWR_{X,Ka}$ is strongly overestimated in "colMix2_Dkernel" and the "CTRL" run. In contrast, $DWR_{X,Ka}$ is well-matched closely above the melting temperature in the "NEW" simulation. The hydrometeor populations with realistic particle sizes are stronger affected by the subsaturated air and sublimate quickly.
Whereas, particles that are too large, sublimate less and retain therefore more mass. Thus the overestimated particle size leads to overestimated precipitation. Between 18:00 UTC and 24:00 UTC, 1.40mm of accumulated rain was observed, 8.91mm simulated by the default simulation and 2.29mm by "colMix2_Akernel". While this represents an overestimates of 536% by the "CTRL" run during this time period, we emphasize the overall good agreement between modeled and observed precipitation





reported by O20 for the entire campaign. While $E_{\text{stick}}$ appeared to be important for the simulated $DWR_{Ka,W}$ in the "snow-

shaft" simulations (Fig. 7), the differences between the simulation with the old ("colMix2_Akernel_LinCot"; Fig. 9(d)) and

the new $E_{\text{stick}}$ parameterization ("colMix2_Akernel"; Fig. 9(c)) are relatively small. In the ICON-LEM simulation, the weaker

growth of the particles in "colMix2_Akernel_LinCot" at lower temperatures might be partly compensated by advection or

nucleation.

Besides the DWRs, MDV provides valuable information about the microphysical properties. As also reported by O20, MDV

is overestimated in the SB06 default simulation especially in regions where the particle sizes are overestimated (Fig. 10). MDV

is often used to distinguish rimed from unrimed particles (e.g., Mosimann, 1995). Using this method, we detect some smaller

episodes where rimed particles are dominating at about 04:00, 18:00, and 22:00 UTC. At other times, the observations indicate

unrimed or only slightly rimed particles. In the SB06 default simulation, high MDVs are obtained in the whole time range

after 18:00. Since the profiles of the hydrometeors show only very little mass of rimed particles during this period, the larger

predicted MDV can be attributed to the overestimation of the unrimed snow particle $v_t$.

The new simulations, all using the new particle properties, have significantly lower values of MDV at all temperatures. This

reduction of MDV compared to the SB06 default setup constitutes a significant reduction of the bias in MDV at temperatures

below -10 °C. For $T >$-10 °C, MDV is even slightly underestimated. Considering that Fig. 5 shows a good agreement of

MDV between the observations and the $v_t$-size relation of "Mix2", we assume that the underestimation of MDV is not caused

by the underestimation of the $v_t$-size relation of aggregates. Since $DWR_{X,Ka}$ also matches well at these temperatures, most

probably other processes than aggregation and sedimentation of unrimed aggregates cause this underestimation of MDV. One

could speculate that riming rates are underestimated or that the vertical air motion is not well simulated.

Most of the findings from the "snowshaft" simulations (e.g. the strong reduction of MDV and DWR at temperatures close to

the melting temperature) are confirmed by the ICON-LEM simulation of this case study. However, the ICON-LEM simulations,

reveal that the influence of $E_{\text{stick}}$ seems to be overestimated in the "snowshaft" simulations. Moreover, accurate modeling of

particle sizes and $v_t$ in the presence of a sublimating layer is critical. The simulations with the new particle properties showed

a slight underestimation of the MDV. This underestimation most likely does not arise from an inaccurate representation of the

particle properties or the aggregation rates but is caused by another process (e.g. riming, vertical air motion). In previous anal-

yses of the SB06 default setup, this underestimation could not have been detected because it was masked by the overestimation

of the aggregate's $v_t$. Because errors can be specific to the chosen day, such as a particular mismatch of the relative humidity,

relying on only one case to detect a discrepancy in the microphysical properties is prone to error. Therefore, we analyze the

statistics of a multi-month simulation in the next section.

## 3.4    Statistical Comparison

After evaluating the choices of the new scheme in the "snowshaft" model and in a case study with ICON-LEM, we perform

ICON-LEM simulations for the entire Tripex time period. By comparing observed and modeled histograms of DWR and MDV

as a function of temperature, we can evaluate the new scheme. Since we additionally contrast the histograms of the "NEW"

and "CTRL" simulations, we can test whether the reduction in the bias of DWRs and MDV found in Sect. 3.3 is specific to the





selected case or rather a consistent feature of the model changes. As DWRs are related to the mean particle size, we can assess

whether the chosen parameter combination can accurately simulate aggregation in various weather situations present in the

simulated days. The same applies to MDV profiles, which are especially valuable in evaluating the suitability of the assumed

$v_t$-size relationship.

The observed and synthetic radar profiles are filtered in the same way for comparability. For example, the first six hours

of simulation and observation are not considered because the model output could contain artifacts during this spin-up time.

Moreover, we include only profiles, in which the rain rate $RR$ exceeds $1\,\mathrm{mmh}^{-1}$. The latter filter enables us to focus on the

most relevant cases for precipitation. Interestingly, O20 found that the discrepancy between model and observations to be

especially obvious for these profiles. For a detailed description of the processing, we refer to O20.

To quantify the agreement between the histograms of the simulations and the observation, the Hellinger distance $H$ is used.

$H$ can be defined for two distributions $P = (p_1, \dots, p_k)$ and $Q = (q_1, \dots, q_k)$ as:

$$H(P,Q) = \frac{1}{\sqrt{2}} \sqrt{\sum_{i=1}^{k} \left(\sqrt{p_i} - \sqrt{q_i}\right)^2} \tag{17}$$

$H$ is zero for two identical distributions and one if the distributions do not overlap at all.

The medians and larger quantiles of the observed distributions of DWRs indicate a strong increase in particle size around -

$15\,^{\circ}\mathrm{C}$ (most evident in $\mathrm{DWR}_{Ka,W}$; Fig. 11(a)) and just above the melting temperature (most evident in $\mathrm{DWR}_{X,Ka}$; Fig. 11(e)).

Both of these characteristic increases of the particle sizes are found to some extent in "CTRL" (panel (b) and (f) in Fig. 11)

and "NEW" (panel (c) and (g) in Fig. 11). The increase of particle sizes between $-15\,^{\circ}\mathrm{C}$ and $-10\,^{\circ}\mathrm{C}$ happens in the new

simulations at slightly lower temperatures and the different profiles reveal a greater variability (visible e.g. in the difference of

$DWR_{Ka,W}$ between the lower and upper decile). $H$ indicates a slightly better match by "CTRL" in this temperature range. For

$T > -10\,^{\circ}\mathrm{C}$ the mean and higher quantiles of $DWR_{X,Ka}$ increase very rapidly in "CTRL" and more slowly in "NEW" and the

observation. The increase of particle sizes as simulated by "NEW" is in much better agreement with the observed profiles. The

upper quartile of $DWR_{X,Ka}$ only slightly exceeds $5\,\mathrm{dB}$ in the observations and "NEW" but is higher than $10\,\mathrm{dB}$ in "CTRL"

for $T > -1\,^{\circ}\mathrm{C}$. This better match is also indicated by $H$ (Fig. 11(h)), which is about five times larger for "CTRL" compared to

"NEW".

Besides the overestimation of $\mathrm{DWR}_{X,Ka}$ closely above the melting temperature, O20 also highlighted the overestimation

of MDV by "CTRL". This overestimation is present at all temperatures (compare panels (a) and (b) in Fig. 12) and can

be attributed to the overestimation of the $v_t$-size relationship of the snow class as reported in Karrer et al. (2020) and the

overestimated particle sizes for the higher temperatures. The overestimation of MDV by "CTRL" is most pronounced for

$T > -15\,^{\circ}\mathrm{C}$. In this temperature range, "CTRL" can not reproduce the asymptotic approach because of the power-law $v_t$-size

relationship (Sect. 3.1.3). For example, the median of $MDV_{Ka}$ at $-5\,^{\circ}\mathrm{C}$ is $1\,\mathrm{ms}^{-1}$ in the observations and at $1.3\,\mathrm{ms}^{-1}$ in

"CTRL". In contrast, the new simulations agree better with the observations and $H$ is about half as large as for "CTRL". The

new scheme setup is more accurate in this temperature range because the Atlas-type $v_t$-size relationship of the "Mix2" particles

(Fig. 3) correctly considers the asymptotic approach to $1\,\mathrm{ms}^{-1}$ at large sizes. However, MDV is slightly underestimated by





"NEW" for $T >$-10 °C. Values substantially above $1 \ \mathrm{ms}^{-1}$ occur in the observations and the new simulations only closely above the melting temperature, where rain is present. At temperatures below -15 °C, both simulations perform similarly, with $H$ being ranging from 0.2 to 0.5. While "CTRL" exhibits a continuous overestimation of MDV, the new simulations lack the observed increase of MDV for $T <$-20 °C. At these temperatures, the selected PSD width (Sect.3.1.1) and cloud ice particle

properties (Sect. 3.1.3) may not be ideal.

The statistical comparison shows that the changes we made to the model could eliminate the most striking biases, namely the overestimation of $DWR_{X,Ka}$ and MDV closely above the melting temperatures. The match of these quantities is important for accurate simulation of precipitation, as exemplified in the case study in Sect. 3.3. Some discrepancies remain, namely the too strong increase of the DWRs at temperatures between -15 °C to -10 °C and the overestimation (underestimation)

of MDV temperatures below -25 °C (above -10 °C). These discrepancies can be caused by several model errors (inaccurate simulation of e.g. PSD shape, $E_{\mathrm{stick}}$, degree of riming, variability in cloud ice properties), that can not be fully deciphered by this observational setup and could benefit from advances in laboratory measurements, observational setup and representation of cloud ice habits and riming degree in the model.

## 4   Conclusions

Aggregation is a key ice microphysical growth process for the formation of precipitating ice particles, which are the precursor of raindrops in cold rain formation. Recent studies using statistics from multi-frequency Doppler radar observations provided observational constraints on how critical radar quantities, such as DWRs or MDV, change with temperature. In this study, we aimed at a deeper analysis of the underlying causes for the observed discrepancies between radar statistics and a state-of-the-art two-moment microphysical scheme and improved its simulation of aggregation.

To this end, as a first step, we revisited all relevant components of aggregation as considered in the two-moment scheme to see how well they represent current knowledge of physics. These components are the size distribution width, the temperature dependence of $E_{\mathrm{stick}}$, the particle properties (with focus on the $v_t$-size relation of aggregates) and the representation of non-spherical particles in the aggregation kernel formulation.

To systematically test the sensitivities of various parameter combinations, we performed 1D simulations with the "snow-

shaft" model, which uses simple profiles of thermodynamic variables and a simple initialization of particles at the model top. Moreover, the model only accounts for a subset of all the microphysical processes that occur in real clouds. Nevertheless, by adjusting the model setup we could match the average profiles of radar observables obtained by the 3D simulations of O20, which used the SB06 default scheme setup.

The "snowshaft" simulations revealed high sensitivity of aggregation to particle properties, the aggregation kernel formu-

lation and $E_{\mathrm{stick}}$. Surprisingly, the size distribution width had a relatively small effect on the modeled mean mass but a considerable influence on the simulated DWRs. The influence of the cloud ice properties was small in both the model and radar variables.


By comparing the profiles from the "snowshaft" simulation with the average observed profiles, we were able to select a set of parameters that provided the best agreement with the observations. In this selection process, we mainly varied the less-known
components e.g. the size distribution width and held constant other parameters that we could better constrain e.g. the $v_t$-size relationship. The size distribution width proved to be a critical component in linking modeled $\bar{x}$ to observed DWRs and at the same time is difficult to constrain with the given observational setup. In particular, we find that the $v_t$-size relationship which accounts for the asymptotic behavior of $v_t$ at large sizes, leads to a better agreement with the observations. Moreover, the A-kernel appears to be a better approximation of the aggregation kernel, when combined with a constant $E_{\mathrm{stick}}$.

We implemented this improved scheme setup in the ICON-LEM model and tested the individual model modifications also in a case study. These more realistic ICON-LEM simulations allowed us to derive potential differences in the analysis of sensitivities compared to the "snowshaft" simulations, possibly caused by effects such as dynamics and advection. Overall, the ICON-LEM simulations yielded similar sensitivities as the "snowshaft" simulations but slight differences were apparent with respect to sensitivity to $E_{\mathrm{stick}}$. The difference between simulations with different $E_{\mathrm{stick}}$ parameterization was less pronounced in
the ICON-LEM simulations. This discrepancy between the simulation frameworks could result from accounting for feedback from microphysics to model humidity in the ICON-LEM simulation.

On the day considered in the case study, relatively dry low-level air resulted in strong sublimation of ice particles. This sublimation feature demonstrated the relevancy of accurately simulating $\bar{x}$. The SB06 default scheme with its largely overestimated aggregate sizes strongly overestimated the rainfall rate on the ground because the large snowflakes could not sublimate fast enough. In contrast, the more realistic aggregate sizes obtained with the new scheme were able to fit the observations much
better.

Finally, the entire period of the campaign dataset (46 days) was simulated again with ICON-LEM using the best matching parameter combination from the previous tests. This allowed us to directly compare the new statistics with the previous analysis of the default scheme provided in O20. The new aggregation formulation is clearly able to reduce the observed overestimation
of MDV. This improvement can be attributed to the new Atlas-type $v_t$-size relationship. The overestimation of the mean particle size at high temperatures revealed in the DWRs was also substantially reduced by the new aggregation parameterization.

Remaining discrepancies are found for DWRs at temperatures of about -12 °C and for MDV at low and high temperatures. The overestimated DWRs by the new simulations could result in overestimated $\bar{x}$ or too broad size distribution in the model. Inclusion of a higher frequency radar (Battaglia et al., 2014) may help to infer the particle growth above -12 °C. The analysis
of Doppler spectra (e.g. similar to Barrett et al., 2019), or observational techniques, e.g. in situ probing of the particle size distribution, would provide additional constraints on the size distribution and ease the interpretation of the MDV and DWR. The mismatch of the MDV at lower temperatures could be caused by an inaccurate size distribution width, as well as $E_{\mathrm{stick}}$ or cloud ice properties. Future studies could focus on this temperature region, which is highly relevant for cloud radiative effects. The slight underestimation of MDV at high temperatures could be due to underestimated riming rates, the representation
of partially rimed particles or other effects as vertical air motion. Further insight could be gained e.g. from the analysis of the Doppler spectra or comparison with other microphysical schemes with a different representation of the riming process (Morrison and Milbrandt, 2015; Tsai and Chen, 2020).





Our approach to investigate the sensitivities of aggregation to the components of its parameterization and improve its simulation in the SB06 microphysics scheme could also be applied to other processes or schemes. Therefore, we summarize the

approach in general terms by the following points:

1. revisit components of the physical parameterization

2. set up single-columns simulations which match the average profiles of simulated observables obtained from long-term 3D simulations with the default scheme setup

3. systematically test the sensitivities of various parameter combinations in 1D simulations

4. select the model configuration that best matches the observations

5. implement model modifications in the 3D model and infer possible differences in sensitivities between 3D simulation and 1D simulations in a Case study

6. rerun the long-term 3D simulation using the best matching parameter combination and investigate the improvements by comparing observations with simulations using the default and the new scheme setup

*Code availability.* The source code of the "snowshaft" model is part of the McSnow model. McSnow and the code to test and optimize the variance approximation of the bulk collision integrals (COLLINT) are hosted on gitlab. The DWD software used in this study (McSnow, COLLINT, etc.) is part of the ICON modeling framework and access can be granted by AS based on an ICON licence agreement. Non-commercial scientific licences for ICON are available at https://code.mpimet.mpg.de/projects/iconpublic/wiki/How_to_obtain_the_model_code

715 .

## Appendix A:  Bulk aggregation rates

We summarize the bulk aggregation formulas for all aggregation processes: ice-snow collection, ice selfcollection, snow self-collection. While the formulations using the D-kernel were already given by Seifert et al. (2014) the formulas using the A-kernel were newly derived in this study.

Combining the definition of the moments,

$$M_n = \int\limits_0^\infty m^n f(m) dm \tag{A1}$$

the SCE (Eq. (4)) and its simplifications in the SB06 scheme (Sect. 3.1), an equation can be derived that allows to calculate all relevant aggregation rates between particles of the classes i and j:

$$\left.\frac{\partial M_{i,n}}{\partial t}\right|_{\mathrm{coll,ij}} = \Phi \int\limits_0^\infty \int\limits_0^\infty f_i(D_i) f_j(D_j) K_{i,j}(D_i, D_j) m_i^n dD_j dD_i \tag{A2}$$





where $M_{j,n}$ is the n-th moment of the hydrometeor class j, $f$ is the particle size distribution for a selected size variable ($D_{\max}$, $D_{eq}$ or $m$), $K$ is the aggregation kernel, and $m$ is the particle mass.

Seifert et al. (2014) uses the variance approach proposed in Seifert and Beheng (2006), which parameterizes the bulk velocity difference by the square root of the second moment of the velocity differences. In this way, the integral is separated into a term containing the geometrical properties ($\mathcal{C}_{n,ij}$) and a part which contains the velocity difference ($\overline{\Delta v}_{n,ij}$) to enable the analytical integration:

$$\left.\frac{\partial M_{i,n}}{\partial t}\right|_{\text{coll,ij}} = \bar{E}_{i,j}\overline{\Delta v}_{n,ij}\mathcal{C}_{n,ij} \tag{A3}$$

The expression of $\mathcal{C}_{n,ij}$ and $\overline{\Delta v}_{n,ij}$ depend on the expression of the PSD$_m$ (Sect. 3.1.1), the formulation of the aggregation kernel (Sect. 3.1.2) and the particle properties (Sect. 3.1.3). The SB06 scheme assumes a modified gamma distribution as a function of mass (Eq. (5)), which can be easily converted to a gamma distribution as a function of $D_{eq}$ if $\mu_m=1/3$ (Eq. (7)). The particle properties are characterized by power-law relations of $m$ (Eq. (12)) and $A_{act}$ (Eq. (15)) vs. $D_{\max}$ and $D_{eq}$. In the new scheme, $v_t$ of cloud ice and snow is parameterized by an Atlas-type relation as a function of $D_{eq}$ (Eq. (11)). Coefficients of the relations can be found in Table 2.

## A1 D-kernel

Inserting the D-kernel (Eq. (1)) into Eq. (A2), the $\mathcal{C}_{n,ij}$ and $\overline{\Delta v}_{n,ij}$ can be written as:

$$\mathcal{C}_{n,ij} = \frac{\pi}{4}\int_0^\infty\int_0^\infty (D_{\max,i}+D_{\max,j})^2 f_i(m_i)f_j(m_j)m_j^n\,dm_i\,dm_j \tag{A4}$$

$$\overline{\Delta v}_{n,ij} = \left\{\frac{1}{\mathcal{N}_{n,ij}}\int_0^\infty\int_0^\infty [v_i(D_{eq,i})-v_j(D_{eq,j})]^2 \times D_{eq,i}^2 D_{eq,j}^2\right.$$

$$\left. f_{eq,i}(D_{eq,i})f_{eq,j}(D_{eq,j})m_i^n\,dD_{eq,i}\,dD_{eq,j}\right\}^{\frac{1}{2}}$$

$$\tag{A5}$$

where $\mathcal{N}$ is the normalization factor given by:

$$\mathcal{N}_{n,ij} = \int_0^\infty\int_0^\infty D_{eq,i}^2 D_{eq,j}^2 f_{eq,i}(D_{eq,i})f_{eq,j}(D_{eq,j})m_i^n\,dD_i\,dD_j \tag{A6}$$

$$\tag{A7}$$

Inserting the $D_{\max}$-$m$ relation:

$$D_{\max,i} = a_i m_i^{b_i} = \frac{\pi\rho_w a_i}{6}D_{eq,i}^{3b_i} \tag{A8}$$

and the PSD$_m$ (Eq. (5)) into $\mathcal{C}_{n,ij}$ (Eq. (A4)) and solving the integral we obtain:

$$\mathcal{C}_{n,ij} = \left(\frac{\pi\rho_w}{6}\right)^n\frac{\pi}{4}N_iN_j\left[\delta_{D,i}^0\bar{D}_i^2+\delta_{D,ij}^n\bar{D}_i\bar{D}_j+\delta_j^n\bar{D}_j^2\right] \tag{A9}$$





, where $\delta_i^n$ and $\delta_j^n$ are equal to $\delta_p^0$ of Eq. (90) of SB2006 and $\delta_{ij}^n$ is equal to $\delta_g^0$ of Eq. (91) of SB2006.

$$\delta_{D,i}^n = \frac{\Gamma((2b_i + \nu_{m,i} + 1 + n)/\mu_{m,i})}{\Gamma((\nu_{m,i} + 1)/\mu_{m,i})} \left[\frac{\Gamma((\nu_{m,i} + 1)/\mu_{m,i})}{\Gamma((\nu_{m,i} + 2)/\mu_{m,i})}\right]^{2b_i + n} \tag{A10}$$

$$\delta_{D,ij}^n = 2\frac{\Gamma((b_i + \nu_{m,i} + 1 + n)/\mu_{m,i})}{\Gamma((\nu_{m,i} + 1)/\mu_{m,i})}\frac{\Gamma((b_j + \nu_{m,j} + 1)/\mu_{m,j})}{\Gamma((\nu_{m,j} + 1)/\mu_{m,j})} \tag{A11}$$

$$\times \left[\frac{\Gamma((\nu_{m,i} + 1)/\mu_{m,i})}{\Gamma((\nu_{m,i} + 2)/\mu_{m,i})}\right]^{b_i + n} \left[\frac{\Gamma((\nu_{m,j} + 1)/\mu_{m,j})}{\Gamma((\nu_{m,j} + 2)/\mu_{m,j})}\right]^{b_j} \tag{A12}$$

Inserting the velocity relation (Eq. (11)) and the size distribution using $D_{eq}$ (Eq. (7)) into the velocity variance (Eq. (A5)) and solving the integral we obtain:

$$\overline{\Delta v}_{n,ij} = \left[(\alpha_{v,j} - \alpha_{v,i})^2 - 2\beta_{v,j}(\alpha_{v,j} - \alpha_{v,i})\left(1 + \frac{\gamma_{v,j}}{\lambda_{eq,j}}\right)^{-\xi_{D,i}^n}\right.$$

$$-2\beta_{v,i}(\alpha_{v,i} - \alpha_{v,j})\left(1 + \frac{\gamma_{v,i}}{\lambda_{eq,i}}\right)^{-\xi_{D,i}^n} + \beta_{v,j}^2\left(1 + \frac{2\gamma_{v,j}}{\lambda_{eq,j}}\right)^{-\xi_{D,j}}$$

$$+\beta_{v,i}^2\left(1 + \frac{2\gamma_{v,i}}{\lambda_{eq,i}}\right)^{-\xi_{D,i}^n} - 2\beta_{v,j}\beta_{v,i}\left(1 + \frac{\gamma_{v,j}}{\lambda_{eq,j}}\right)^{-\xi_{D,j}}$$

$$\left. \times \left(1 + \frac{\gamma_{v,i}}{\lambda_{eq,i}}\right)^{-\xi_{D,i}^n}\right]^{\frac{1}{2}} \tag{A13}$$

with

$$\xi_{D,i}^n = \mu_{eq,i} + 3 + 3n \tag{A14}$$

$$\xi_{D,j} = \mu_{eq,j} + 3 \tag{A15}$$

## A2 A-kernel

Inserting the A-kernel (Eq. (9)) into Eq. (A2), the velocity variance and the geometric part of the bulk collision rates can be written as:

$$\mathcal{C}_{n,ij} = \int_0^\infty \int_0^\infty \left(A_i^{0.5} + A_j^{0.5}\right)^2 f_i(D_i)f_j(D_j)m_j^n dD_i dD_j \tag{A16}$$

$$\overline{\Delta v}_{n,ij} = \left\{\frac{1}{\mathcal{N}_{n,ij}}\int_0^\infty \int_0^\infty [v_i(D_{eq,i}) - v_j(D_{eq,j})]^2 \times D_{eq,i}^{\sigma_{A,i}} D_{eq,j}^{\sigma_{A,j}}\right.$$

$$\left. f_{eq,i}(D_{eq,i})f_{eq,j}(D_{eq,j})m_i^n dD_{eq,i}dD_{eq,j}\right\}^{\frac{1}{2}}$$

$$\mathcal{N}_{n,ij} = \int_0^\infty \int_0^\infty D_{eq,i}^{\sigma_{A,i}} D_{eq,j}^{\sigma_{A,j}} f_{eq,i}(D_{eq,i})f_{eq,j}(D_{eq,j})m_i^n dD_{eq,i}dD_{eq,j} \tag{A17}$$

$$\tag{A18}$$





Inserting the $A$-$D_{\text{eq}}$ relation (Eq. (15)) and the size distribution as a function of $D_{\text{eq}}$ (Eq. (7)) into the geometric part (Eq. (A16)) and solving the integral leads to:

$$
\quad \mathcal{C}_{n,ij} = \left(\frac{\pi\rho_w}{6}\right)^n N_i N_j \left[\delta^n_{A,i}\bar{D}^{\sigma^*_{A,i}}_{\text{max},i} + \delta^n_{A,ij}\bar{D}^{\sigma^*_{A,i}/2}_{\text{max},i}\bar{D}^{\sigma^*_{A,j}/2}_{\text{max},j} + \delta^n_{A,j}\bar{D}^{\sigma^*_{A,j}}_{\text{max},j}\right]
$$
(A19)

with:

$$
\delta^n_{A,i} = \gamma_{A,i}\frac{\Gamma(\mu_{\text{eq},i}+\sigma_{A,i}+1+3n)}{\Gamma(\mu_{\text{eq},i}+1)}c^{\sigma_{A,i}+3n}_{\lambda,i}
$$
(A20)

$$
\delta^n_{A,ij} = 2(\gamma_{A,i}\gamma_{A,j})^{0.5}\frac{\Gamma(\mu_{\text{eq},i}+\sigma_{A,i}/2+1+3n)}{\Gamma(\mu_{\text{eq},i}+1)}c^{\sigma_{A,i}/2+3n}_{\lambda,i}
$$
(A21)

$$
\qquad \times\frac{\Gamma(\mu_{\text{eq},j}+\sigma_{A,j}/2+1)}{\Gamma(\mu_{\text{eq},j}+1)}c^{\sigma_{A,j}/2}_{\lambda,j}
$$

$$
\delta^n_{A,j} = \gamma_{A,j}\frac{\Gamma(\mu_{\text{eq},j}+\sigma_{A,j}+1)}{\Gamma(\mu_{\text{eq},j}+1)}c^{\sigma_{A,j}}_{\lambda,j}
$$
(A22)

$$
\qquad \times\left[\frac{\Gamma(\mu_{\text{eq},i}+4)}{\Gamma(\mu_{\text{eq},i}+1)}\right]^n c^{3n}_{\lambda,i}
$$

$$
\sigma^*_{A,i} = \frac{b_{m,i}\sigma_{A,i}}{3}
$$
(A23)

$$
c_{\lambda,i} = \left[\frac{6a_{m,i}}{\pi\rho_w}\frac{\Gamma(\mu_{\text{eq},i}+1)}{\Gamma(\mu_{\text{eq},i}+4)}\right]^{1/3}
$$
(A24)

Inserting the velocity relation (Eq. (11)) and the size distribution as a function of $D_{\text{eq}}$ (Eq. (7)) into the velocity variance (Eq. (A17)) and solving the integral we obtain:

$$
\overline{\Delta v}_{n,ij} = \left[(\alpha_{v,j}-\alpha_{v,i})^2 - 2\beta_{v,j}(\alpha_{v,j}-\alpha_{v,i})\left(1+\frac{\gamma_{v,j}}{\lambda_{\text{eq},j}}\right)^{-\xi_{A,j}}\right.
$$

$$
\qquad -2\beta_{v,i}(\alpha_{v,i}-\alpha_{v,j})\left(1+\frac{\gamma_{v,i}}{\lambda_{\text{eq},i}}\right)^{-\xi^n_{A,i}} + \beta^2_{v,j}\left(1+\frac{2\gamma_{v,j}}{\lambda_{\text{eq},j}}\right)^{-\xi_{A,j}}
$$

$$
\qquad +\beta^2_{v,i}\left(1+\frac{2\gamma_{v,i}}{\lambda_{\text{eq},i}}\right)^{-\xi^n_{A,i}} - 2\beta_{v,j}\beta_{v,i}\left(1+\frac{\gamma_{v,j}}{\lambda_{\text{eq},j}}\right)^{-\xi_{A,j}}
$$

$$
\qquad \left.\times\left(1+\frac{\gamma_{v,i}}{\lambda_{\text{eq},i}}\right)^{-\xi^n_{A,i}}\right]^{\frac{1}{2}}
$$
(A25)

with

$$
\xi^n_{i,A} = \mu_{\text{eq},i}+\sigma_{A,i}+1+3n
$$
(A26)

$$
\xi_{j,A} = \mu_{\text{eq},j}+\sigma_{A,j}+1
$$
(A27)

7979





## A3 Ice selfcollection

### A3.1 D-kernel

For ice selfcollection the geometry part (Eq. (A16)) simplifies to:

$$\mathcal{C}_{n,ii} = \left(\frac{\pi\rho_w}{6}\right)^n \frac{\pi}{4} N_i^2 \left[2\delta_{D,i}^0 + \delta_{D,ii}^n\right] \bar{D}_i^2 \tag{A28}$$

, where $\delta_i^n$ is equal to $\delta_p^0$ of Eq. (90) of SB2006 and $\delta_{ii}^n$ is equal to $\delta_g^0$ of Eq. (91) of SB2006.

$$\delta_{D,i}^n = \frac{\Gamma((2b_i+\nu_{m,i}+1+n)/\mu_{m,i})}{\Gamma((\nu_{m,i}+1)/\mu_{m,i})} \left[\frac{\Gamma((\nu_{m,i}+1)/\mu_{m,i})}{\Gamma((\nu_{m,i}+2)/\mu_{m,i})}\right]^{2b_i+n} \tag{A29}$$

$$\delta_{D,ii}^n = 2\frac{\Gamma((b_i+\nu_{m,i}+1+n)/\mu_{m,i})}{\Gamma((\nu_{m,i}+1)/\mu_{m,i})^2}\Gamma((b_i+\nu_{m,i}+1)/\mu_{m,i})$$

$$\times \left[\frac{\Gamma((\nu_{m,i}+1)/\mu_{m,i})}{\Gamma((\nu_{m,i}+2)/\mu_{m,i})}\right]^{2b_i+n} \tag{A30}$$

The velocity variance simplifies to:

$$\overline{\Delta v}_{n,ii} = \beta_{v,i}\sqrt{2}\left[\left(1+\frac{2\gamma_{v,i}}{\lambda_{eq,i}}\right)^{-\xi_{D,i}^n} - \left(1+\frac{\gamma_{v,i}}{\lambda_{eq,i}}\right)^{-2\xi_{D,i}^n}\right]^{\frac{1}{2}} \tag{A31}$$

with

$$\xi_{D,i}^n = \mu_{eq,i}+3+3n \tag{A32}$$

### A3.2 A-kernel

For ice selfcollection $\mathcal{C}$ (Eq. (A16)) simplifies to:

$$\mathcal{C}_{n,ii} = \left(\frac{\pi\rho_w}{6}\right)^n N_i^2 \left[\delta_{A,i}^n + \delta_{A,ii}^n + \delta_{A,i2}^n\right] \left(\frac{6a_{m,i}}{\pi\rho_w}\right)^{\frac{\sigma_{A,i}}{3}} \min\left(\gamma_{A,i}\left(\frac{\pi\rho_w}{6a_{m,i}}\right)^{\frac{\sigma_{A,i}}{3}}\bar{D}_{max,i}^{\sigma_{A,i}}, \frac{\pi}{4}\bar{D}_{max,i}^2\right) \tag{A33}$$

with:

$$\delta_{A,i}^n = \frac{\Gamma(\mu_{eq,i}+\sigma_{A,i}+1+3n)}{\Gamma(\mu_{eq,i}+1)}c_{\lambda,i}^{\sigma_{A,i}+3n} \tag{A34}$$

$$\delta_{A,ii}^n = 2\frac{\Gamma(\mu_{eq,i}+\sigma_{A,i}/2+1+3n)}{\Gamma(\mu_{eq,i}+1)^2}\Gamma(\mu_{eq,i}+\sigma_{A,i}/2+1)c_{\lambda,i}^{\sigma_{A,i}+3n} \tag{A35}$$

$$\delta_{A,i2}^n = \frac{\Gamma(\mu_{eq,i}+\sigma_{A,i}+1)}{\Gamma(\mu_{eq,i}+1)}c_{\lambda,i}^{\sigma_{A,i}+3n}\left[\frac{\Gamma(\mu_{eq,i}+4)}{\Gamma(\mu_{eq,i}+1)}\right]^n \tag{A36}$$

$$\sigma_{A,i}^* = \frac{b_{m,i}\sigma_{A,i}}{3} \tag{A37}$$

$$c_{\lambda,i} = \left[\frac{6a_{m,i}}{\pi\rho_w}\frac{\Gamma(\mu_{eq,i}+1)}{\Gamma(\mu_{eq,i}+4)}\right]^{1/3} \tag{A38}$$



For small sizes, the parametrization of $A_{act}$ yields values of $A_r$ larger than one (e.g., columns smaller than 8e-5 m; Fig. 3(d)). For small mean sizes, these particles with unphysical $A_r$ can contribute substantially to $\mathcal{C}_{n,ii}$. Therefore, we limit $A_{act}$ to $A_{sphere}$ in Eq. (A33). The effect of this limiter can be seen in the kink of the bulk collision rates (Fig. A3(c) and (d)).

Inserting the velocity relation (Eq. (11)) and the size distribution using $D_{eq}$ (Eq. (7)) into the velocity variance (Eq. (A17)) and solving the integral we find:

$$\overline{\Delta v}_{n,ii} = \sqrt{2}\beta_{v,i}\left[\left(1+\frac{2\gamma_{v,i}}{\lambda_{\text{eq},i}}\right)^{-\xi^n_{A,i}} - \left(1+\frac{\gamma_{v,i}}{\lambda_{\text{eq},i}}\right)^{-2\xi^n_{A,i}}\right]^{\frac{1}{2}} \tag{A39}$$

with

$$\xi^n_{i,A} = \mu_{\text{eq},i} + \sigma_{A,i} + 1 + 3n \tag{A40}$$

$$\tag{A41}$$

### A4    Snow selfcollection

#### A4.1    D-kernel

For snow selfcollection only the first moment is relevant and $\mathcal{C}$ (Eq. (A16)) simplifies to:

$$\mathcal{C}_{0,ss} = \frac{\pi}{4}N_s^2\left[2\delta^0_{D,s} + \delta^0_{D,ss}\right]\bar{D}_s^2 \tag{A42}$$

, where $\delta^n_s$ is equal to $\delta^0_p$ of Eq. (90) of SB2006 and $\delta^n_{ss}$ is equal to $\delta^0_g$ of Eq. (91) of SB2006.

$$\delta^0_{D,s} = \frac{\Gamma((2b_s+\nu_{m,s}+1)/\mu_{m,s})}{\Gamma((\nu_{m,s}+1)/\mu_{m,s})}\left[\frac{\Gamma((\nu_{m,s}+1)/\mu_{m,s})}{\Gamma((\nu_{m,s}+2)/\mu_{m,s})}\right]^{2b_s} \tag{A43}$$

$$\delta^0_{D,ss} = 2\left[\frac{\Gamma((b_s+\nu_{m,s}+1)/\mu_{m,s})}{\Gamma((\nu_{m,s}+1)/\mu_{m,s})}\right]^2\left[\frac{\Gamma((\nu_{m,s}+1)/\mu_{m,s})}{\Gamma((\nu_{m,s}+2)/\mu_{m,s})}\right]^{2b_s} \tag{A44}$$

     The velocity variance simplifies to:

$$\overline{\Delta v}_{0,ss} = \sqrt{2}\beta_{v,s}\left[\left(1+\frac{2\gamma_{v,s}}{\lambda_{\text{eq},s}}\right)^{-\xi_{D,s}} - \left(1+\frac{\gamma_{v,s}}{\lambda_{\text{eq},s}}\right)^{-2\xi_{D,s}}\right]^{\frac{1}{2}} \tag{A45}$$

with

$$\xi_{D,s} = \mu_{\text{eq},s} + 3 \tag{A46}$$

$$\tag{A47}$$

#### A4.2    A-kernel

$\mathcal{C}$ of the A-kernel simplifies for snow selfcollection to:





$$\mathcal{C}_{0,ss} = N_s^2 \left[ 2\delta_{A,s}^0 + \delta_{A,ss}^0 \right] \left( \frac{6a_{m,s}}{\pi\rho_w} \right)^{\frac{\sigma_{A,s}}{3}} \min\left( \gamma_{A,s} \left( \frac{\pi\rho_w}{6a_{m,s}} \right)^{\frac{\sigma_{A,s}}{3}} \bar{D}_{\max,s}^{\sigma_{A,s}^*}, \frac{\pi}{4} \bar{D}_{\max,s}^2 \right)$$

$$(A48)$$

with:

$$\delta_{A,s}^0 = \frac{\Gamma(\mu_{\mathrm{eq},s} + \sigma_{A,s} + 1)}{\Gamma(\mu_{\mathrm{eq},s} + 1)} c_{\lambda,s}^{\sigma_{A,s}} \tag{A49}$$

$$\delta_{A,ss}^0 = 2\frac{\Gamma(\mu_{\mathrm{eq},s} + \sigma_{A,s}/2 + 1)^2}{\Gamma(\mu_{\mathrm{eq},s} + 1)^2} c_{\lambda,s}^{\sigma_{A,s}}$$

$$\sigma_{A,s}^* = \frac{b_{m,s}\sigma_{A,s}}{3} \tag{A50}$$

$$c_{\lambda,s} = \left[ \frac{6a_{m,s}}{\pi\rho_w} \frac{\Gamma(\mu_{\mathrm{eq},s} + 1)}{\Gamma(\mu_{\mathrm{eq},s} + 4)} \right]^{1/3} \tag{A51}$$

The area ratios are limited in the same way as for ice selfcollection.

The velocity variance simplifies to:

$$\overline{\Delta v}_{0,s} = \sqrt{2}\beta_{v,s} \left[ \left( 1 + \frac{2\gamma_{v,s}}{\lambda_{\mathrm{eq},s}} \right)^{-\xi_{A,s}} - \left( 1 + \frac{\gamma_{v,s}}{\lambda_{\mathrm{eq},s}} \right)^{-2\xi_{A,s}} \right]^{0.5} \tag{A52}$$

$$850 \tag{A53}$$

with

$$\xi_{A,s} = \mu_{\mathrm{eq},s} + \sigma_{A,s} + 1 \tag{A54}$$

$$(A55)$$

## Appendix B: Atmospheric Setup for 1D-simulation and Atmospheric Fields of the Case Study Predicted by
**ICON-LEM**

Fig. A5 shows the atmospheric variables from O20 simulations and the setup for the "snowshaft" simulations. Fig. A6 shows the atmospheric variables of the case study.

*Author contributions.* MK performed and analyzed the "snowshaft" and ICON-LEM simulation. MK and AS derived the new bulk aggregation formulations. MK and DO performed the forward operation and analyzed the statistics of the multi-month ICON-LEM simulations. SK
supervised the project. MK and SK prepared the paper with contributions from all co-authors.

*Competing interests.* The authors declare that they have no conflicts of interest.





| collision partners | | | | | |
| i | j | $\frac{\partial N_{cloudice}}{\partial t}$ | $\frac{\partial L_{cloudice}}{\partial t}$ | $\frac{\partial N_{snow}}{\partial t}$ | $\frac{\partial L_{snow}}{\partial t}$ |
| --- | --- | --- | --- | --- | --- |
| cloud ice | cloud ice | -1 | -1 | +1/2 | +1 |
| cloud ice | snow | -1 | -1 | 0 | +1 |
| snow | snow | 0 | 0 | -1 | 0 |

**Table A1.** Prefactor Φ of the aggregation rates (Eq. (A2) for different aggregation processes and the predicted moments of the cloud ice and snow distribution.

*Acknowledgements.* Contributions by MK, SK and DO were funded by the German Research Foundation (DFG) under grant KN 1112/2-1 as part of the Emmy-Noether Group "Optimal combination of Polarimetric and Triple Frequency radar techniques for Improving Microphysical process understanding of cold clouds" (OPTIMIce). AS received funding from the German Science Foundation (DFG) under grant SE 1784/3-1, project ID 408011764 (IMPRINT) as part of the DFG priority program SPP 2115 (PROM) on radar polarimetry. MK also acknowledges support by the Graduate School of Geosciences of the University of Cologne. We thank Vera Schemann for the support with the ICON-LEM simulations. This work used resources of the Deutsches Klimarechenzentrum (DKRZ) granted by its Scientific Steering Committee (WLA) under project ID bb1086.






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

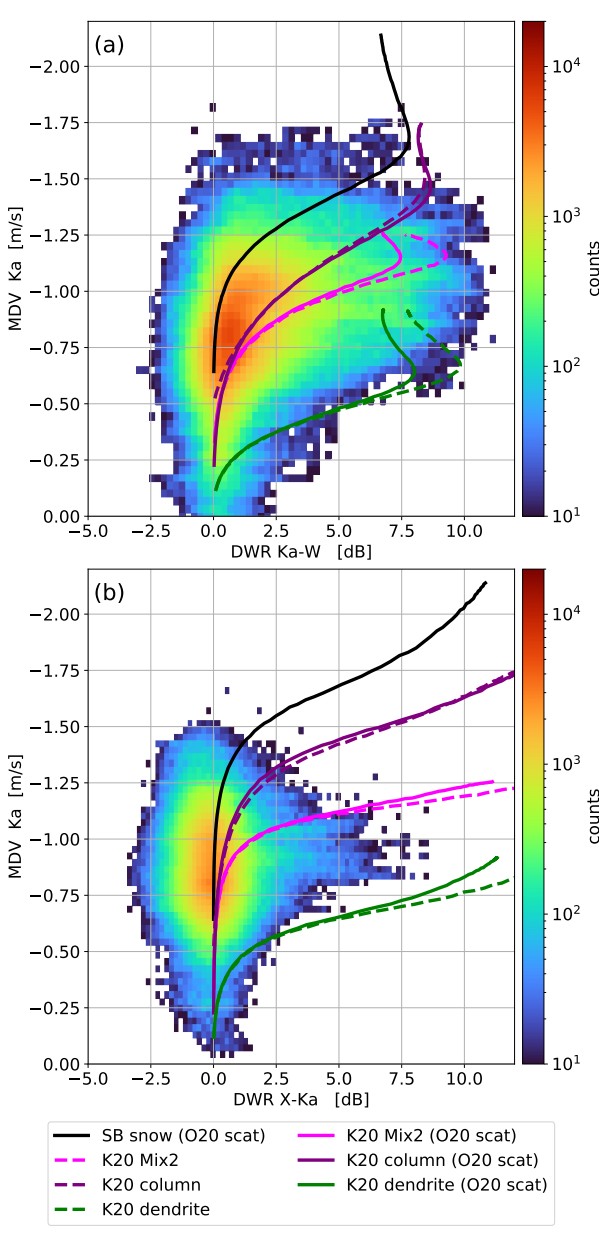

**Figure 5.** Comparison of the modelled and observed relationship between MDV and DWR ((a) $DWR_{Ka,W}$, (b) $DWR_{X,Ka}$). The histogram shows the observations from the Tripex campaign (D18). The lines show the theoretical MDV at a given DWR for the $v_t$ size relations of snow particles as assumed in the SB default (black) and as modeled in K20. For the dashed lines, the SSRGA parameters have been directly derived from the corresponding aggregate ensemble properties (as found in Ori et al. (2020b)). The solid lines use SSRGA parameters as used in O20 in order to illustrate the uncertainty due to the scattering parameters. The lines are calculated using PAMTRA and the properties of the US standard winter atmosphere at 700hPa.

**Figure 6.** Profiles of model variables in the "snowshaft" simulations. Number density $N$ (left), mass mixing ratio $Q$ (middle) and mean mass $\bar{x}$ (right) of the cloud ice (top), snow (middle) and the sum of cloud ice and snow (bottom). Lines: Simulations using different model settings as described in Table 3. Greyscale: Histogram of the hydrometeor contents vs. temperature from the ICON-LEM simulations of the Tripex campaign O20 filtered to include only profiles where the precipitation rate exceeds 1mm/h. The simulations in O20 used the default model settings.

.

**Figure 7.** a) Reflectivity $Ze_{Ka}$, b) Mean Doppler velocity $MDV_{Ka}$, c) $DWR_{Ka,W}$ d) $DWR_{X,Ka}$. Lines: Simulated profiles based on "snow-shaft" simulations (Fig. 6) and median and quartiles of the observations. Greyscale: Histogram of observations from the Tripex campaign (O20).



**Figure 8.** Time-height profile of $Ze_{Ka}$ from 3rd January 2016 as observed (a) and simulated (b-g) with various model settings (Table 3) . Shown are also selected temperature isolines from CloudNet (Illingworth et al., 2007) for the observations (a) and the corresponding ICON-LEM model output (b-g).





**Figure 9.** Same as Fig. 8 but displaying $DWR_{X,Ka}$, which is sensitive to mean mass diameters of 1 to 20 mm





**Figure 10.** Same as Fig. 8 but displaying $MDV_{Ka}$, which is strongly linked to $v_t$



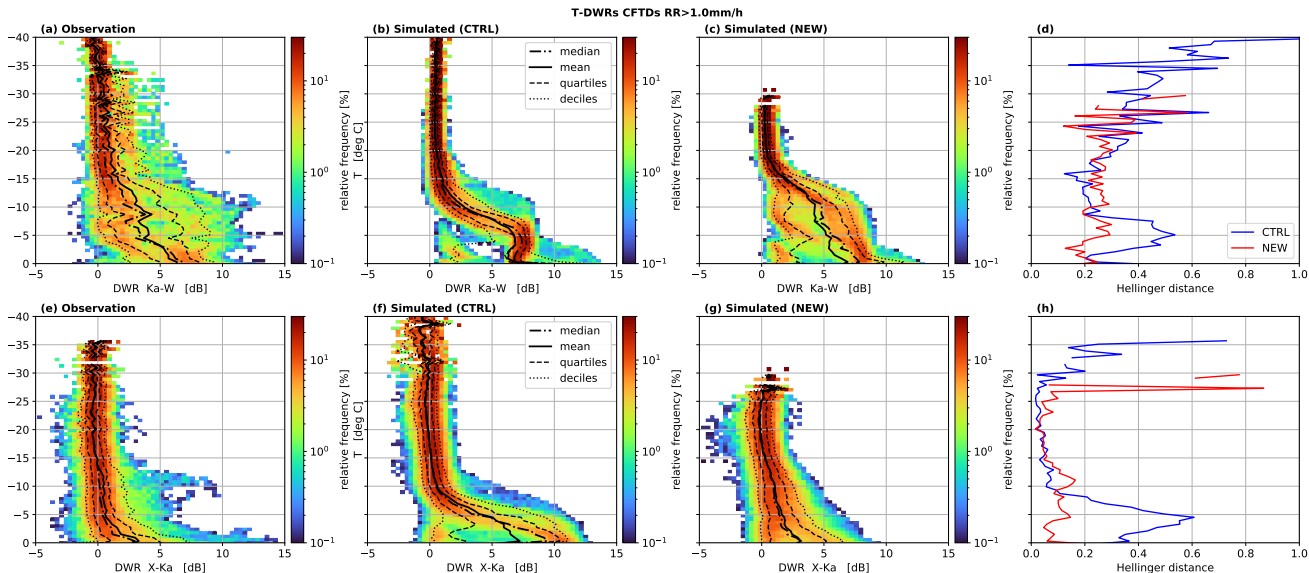

**Figure 11.** Contour frequency by temperature diagrams (CFTDs) for all profiles with RR>1 $mmh^{-1}$ of the dual-wavelength ratios between X- and Ka-Band ($DWR_{X,Ka}$, top) and Ka- and W-Band ($DWR_{X,Ka}$, bottom) from the default simulation (a and e), the new simulation (colMix2_Akernel; b and f) and measured (c and g). The black lines represent the statistical measures (median, mean, quartiles, and deciles) at different temperatures. Panel d and h show the Hellinger distance between the simulated and observed distributions for all temperatures.





**Figure 12.** Contour frequency by temperature diagrams (CFTDs) of mean Doppler velocity of the Ka-Band ($MDV_{Ka}$) from the default simulation (a and e), the new simulation (colMix2_Akernel, b and f), and measured (c and g). The black lines represent the statistical measures (median, mean, quartiles and deciles) at different temperatures. The histograms on top are calculated including all data, on the bottom only data from profiles where the precipitation rate RR exceeds 1mm/h. A vertical line at 1m/s eases the comparison of the different distributions. Panel d and h show the Hellinger distance between the simulated and observed distributions for all temperatures.



**Figure A1.** Numeric and analytic solution of the bulk collision rates ((a) and (c): normalized number collision rates) of ice(column)-snow(Mix2) collisions for Atlas-type and power-law velocity size relations and D- ((a) and (b)) and A-kernel ((c) and (d)) formulations. Shape parameter is $\mu_{eq}=2$ (Eq. (7)), which is equal to $\mu_m=0$ (Eq. (5)) for cloud ice and snow. Left: number density, right: mass density; top: D-kernel; bottom: A-kernel



**Figure A2.** Same as Fig. A1 but with $\mu_{eq}=8$ (Eq. (7)), which is equal to $\mu_m=2$ (Eq. (5)) for snow.

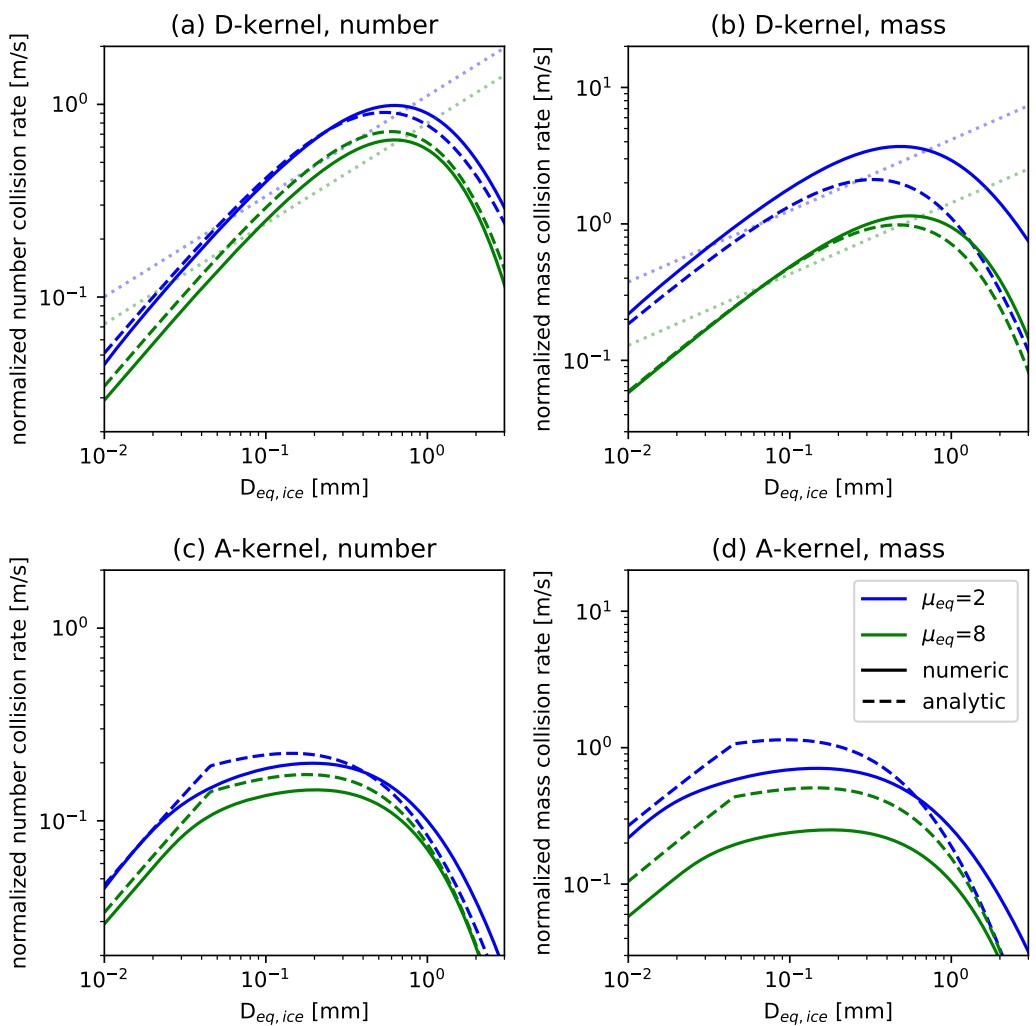

**Figure A3.** Numeric and analytic solution of the bulk collision rates of ice(column)-ice(column) collisions ((a) and (c): number, (b) and (d): mass; (a) and (b): D-kernel, (c) and (d): A-kernel).

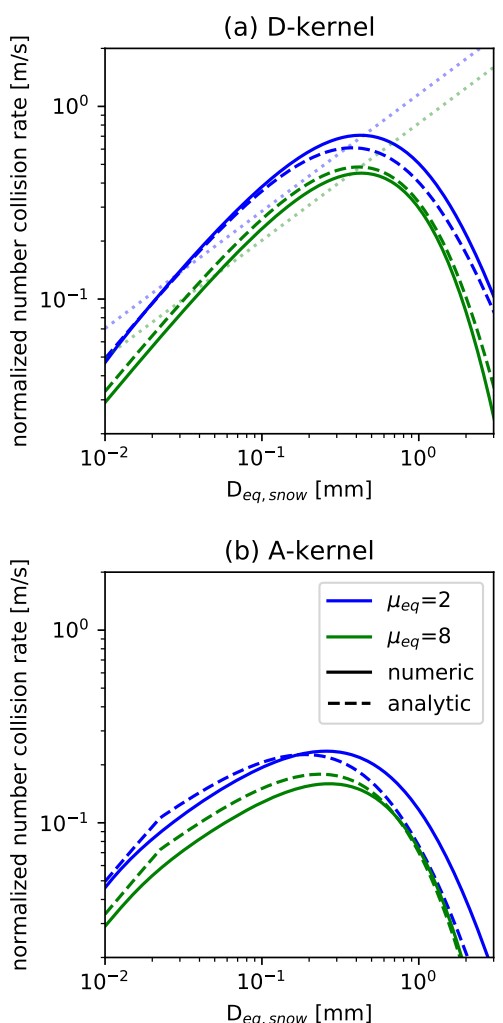

**Figure A4.** Numeric and analytic solution of the bulk collision rates of snow(Mix2)-snow(Mix2) collisions (top: D-kernel; bottom: A-kernel)



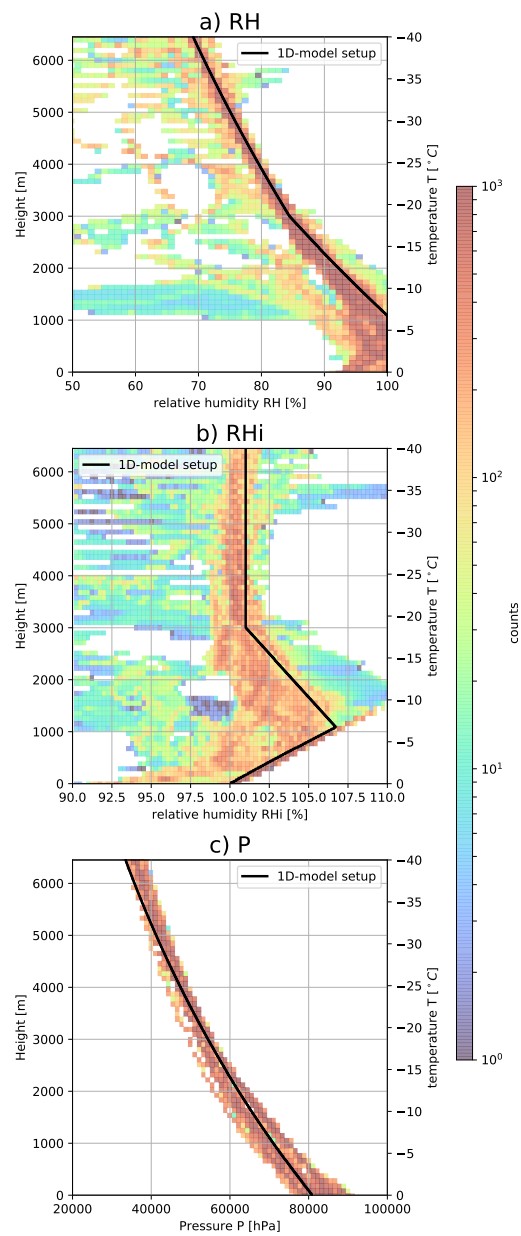

**Figure A5.** Setup of atmospheric variables in the 1D-simulations (Sect. 3.2) (black line) which was chosen based on the histograms from the ICON-LEM simulation done in (histogram is shown in the background, O20). The histogram is filtered to include only profiles where the rain rate exceeds 1mm/h. (a) temperature, (b) relative humidity with respect to water, (c) relative humidity with respect to ice. The height of the melting temperature $0°C$ is set to 0m and other heights are calculated assuming a temperature gradient of 0.0062K/m. Color-coded are the counts in the ICON-LEM simulations from O20.



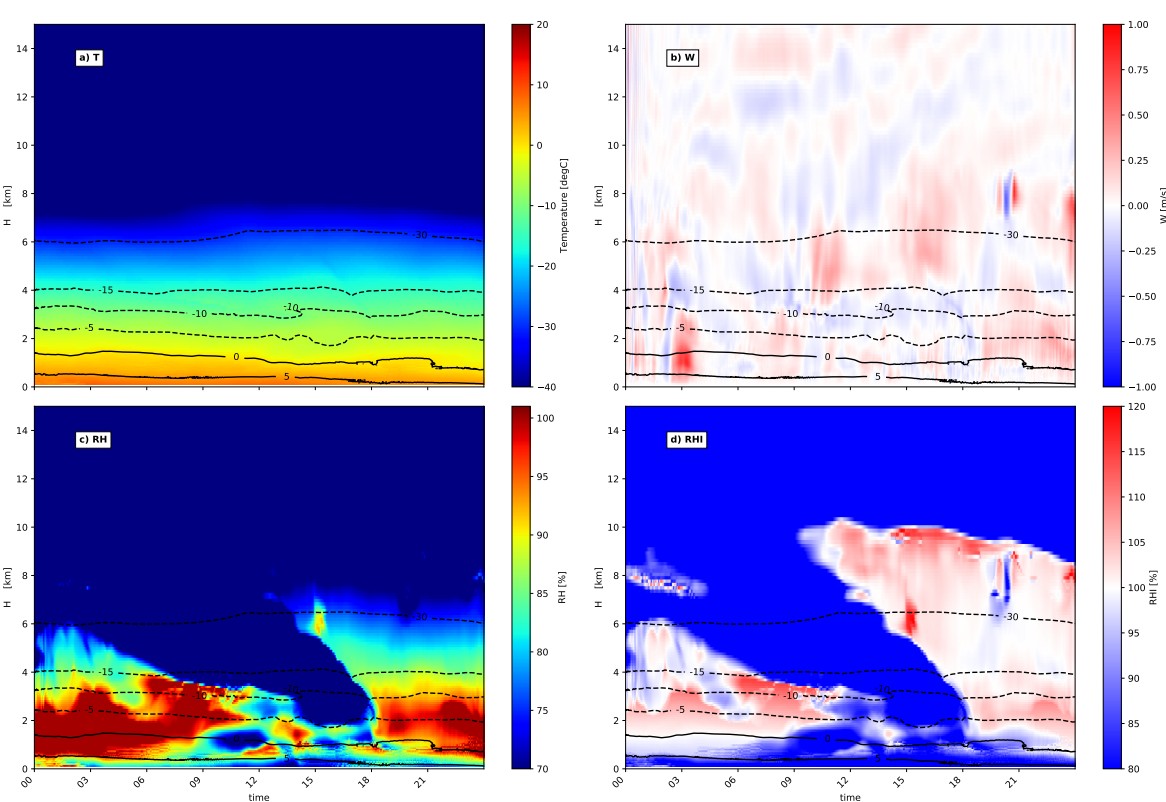

**Figure A6.** Temperature (a), vertical velocity (b), relative humidity with respect to water (c) and ice d) over Juelich in the SB06 default simulation on the January 3, 2016. Temperature isolines are shown on each plot.