# Peer review of "Improving the Representation of Aggregation in a Two-moment Microphysical Scheme with Statistics of Multi-frequency Doppler Radar Observations"

_Atmospheric Chemistry and Physics, 2021_

## Referee Comment (RC2)

**Review of "Improving the Representation of Aggregation in a Two-moment Microphysical Scheme with Statistics of Multi-frequency Doppler Radar Observations"**

**General Comments:**

This paper is concerned with the representation of ice particle aggregation in a bulk microphysics scheme. The paper uses observations from multifrequency and doppler radar to revise the necessary parameters. The methods seem to be very detailed and well-justified. Perhaps the weakest point is that the model used is 1-D and tuned to the observations, but these shortcomings are recognised and a reasonably well-described method for tuning the 1-D model is described.

Overall, I feel that the paper is an excellent contribution to the literature. It is a very dense paper, and I felt that sometimes the main message was lost a little in the text. Perhaps some of the technical detail (e.g. line 609 about the statistical metric used) could have been moved to the appendix and the main findings stated in the abstract? – at the moment this is more general findings that are stated.

I have no strong arguments against the paper, and recommend publication after considering some minor points.

**Specific Comments:**

Line 33: I thought the sentence was confusing to the uninformed reader about the fall-speed of ice. This could be easily reworded to make it less ambiguous.

Line 56: Atlas-type vt-size relation? Wasn't clear what this meant at first, but I see this is defined on page 13, line 322. Maybe this could be stated sooner.

Section 2.3: should rho_air be included? Mass mixing ratio is kg/kg. Assuming that f(m) is number per kg per mass interval, this would mean the units of Q are kg/m-3, which is ice water content.

Section 2.4, line 205: DWR is calibrated to remove attenuation using disdrometer measurements. Here the authors argue that DWR is correlated to the mean mass of the distribution. I did not understand the arguments why this is the case. Could this be made clearer?

---

## Author Comment (AC1)

**Reviewer #1:**

In this study the authors use detailed multi-frequency radar observations in order to constrain key parameters in a 2-moment bulk microphysics scheme that are important for the parameterization of snow aggregation. The authors examine the problem in detail in a simple 1D context and then expand their tests using a 3D LES model. Overall, I think this is a great paper and a solid piece of scientific work. I liked the authors' initial premise that tuning of physics parameterizations based on "large-scale" results can be deceiving due to the potential for compensating model errors and their strategy of attacking the problem on an observation-based process level study. This paper is an excellent illustration of how to tune (constrain) a microphysics scheme on a process level – which is a difficult task – using observations.

I really do not have any constructive comments to add regarding things that could be improved in the paper, which I think is essentially publishable in its current form. The comments I made below are simply offered as food for thought for the authors, which they may wish to comment in the paper (as they see fit). Overall, great paper.

**General comment by the authors to Reviewer #1:**

We thank the reviewer for the time and effort in reading the manuscript and providing constructive comments and suggestions. In the following, we address these comments and suggestions point by point. Line numbers refer to the non-revised manuscript.

**Point-by-point reply of the authors to the questions and comments raised by reviewer#1**

Specific Comments

This study makes comparisons between direct radar measurements (i.e. variables in radar space) to comparable model variables that are computed using instrument simulators. If I understand correctly, the alternative

approach would be to apply retrievals to the radar observations and convert those to fields that are more directly comparable to model fields. Could you please comment on the relative strengths and weaknesses of the two approaches? For modelers, the second approach seems more intuitive. (I may be getting confused with dual-polarization retrievals.)

**A: We believe that both retrieval (observation-to-model approach) and forward operation (model-to-observation approach) have their advantages. Retrievals make the comparison in model space possible, which might be more intuitive, as the reviewer suggests.**

**The main advantages of forward operation might be that uncertainties can be evaluated in the observational space and it might be easier to trace uncertainties arising from the assumptions which have to be made in the forward operator.**

**An example, that illustrates the advantage of considering uncertainties in the observational space is the interpretation of the biases between model and observation in Figure 11. Although at lower temperatures only small crystal are present, DWRs have a spread of about 1-2dB, which can be due to natural variability and measurement uncertainty. This spread of DWR can be used to asses which deviations are significant and which should not be overinterpreted.**

**To remind the reader of the motivation why we use the forward operation approach given in Ori et al. 2020, we included some additional sentences at line 81:**

**"The comparison of model and observations in the radar space using a radar forward operator simplifies the assessment of uncertainties because the deviations between model and observation can be directly compared to the variability of the observation.**
**The alternative approach of applying a retrieval to the observations might seem more intuitive because microphysical variables, such as number density, can be compared directly. However, assuring consistency between model and retrieval as well as tracing the propagation of uncertainties, for example in the observables or the forward model, is often more complicated to obtain (e.g., Reitter, 2011)."**

Could (and should) the approach used in this study be applied to parameterize tendency rates for the spectral width (i.e. for triple-moment treatment of snow) for aggregation (and other processes, such as break-up)? Could you please comment on whether triple-moment snow would improve the representation of the effects of these processes?

A three-moment representation of the snow could be advantageous since the width of the size distribution (spectral width) is a critical parameter for linking the modeled mean mass to the observed dual-wavelength ratios (DWRs). However, the observables considered (reflectivity, DWRs, mean Doppler velocity) mainly constrain the mean values of the particle population, such as integrated mass, mean size, and mean velocity. In contrast, these observables contain only limited constraints on the variance of the particle population, such as size. Therefore, we argue that additional observables from radar (e.g., Doppler spectrum width) or in situ observations should be considered when applying the approach with a three-moment microphysics scheme.

We added the following sentences to the discussion in line 668:

"Therefore, using a microphysical scheme that explicitly simulates the width of the size distribution (e.g., a three-moment scheme) would provide a more consistent link between model and observation. However, additional observational constraints from radar (Doppler spectral width) and in situ observations should be considered in this case."

> In the collection kernels, is there not a slight "break-down" for the situation where the collector and collectee particles have similar sizes, and hence fall speeds, for the analytic solutions (that is, is the kernel values are underestimated)? It was my understanding that that was one of the reasons some schemes use numerical calculations and lookup tables to compute collection rates. On the other hand, I guess the good results summarized in Fig. A1 speak for themselves.

A: In contrast to the Wisner approximation, collision rates are non-zero in the variance approximation of Seifert and Beheng (2006) also if the bulk velocities of the collector and the collectee are the same (this also includes selfcollection). Fig. A1, indeed, demonstrates that the variance approximation deviates only little from the numerical solution, even in this difficult case of similar bulk velocities. However, one can easily imagine that assumptions about the size distribution widths are especially critical in this case. We added the following sentence to motivate better the approximation of Seifert and Beheng (2006) given in Appendix A already in the main text at line 180.

"The latter is made possible by considering the square root of the second moment of the velocity differences, which also has the advantage over the approximation by Wisner (1972) that the collision rates between different particles are non-zero even if their bulk velocities are equal."

Minor Points

1. Line 139 and beyond: Probably at this point you could stop using quotation marks when writing "snowshaft".

**A: Done. Thanks for the comment.**

2. Line 160: "horizontal resolution" should be "horizontal grid spacing"; "vertical resolution" should be "vertical grid spacing".

**A: Corrected.**

3. Line 293: On the other hand, the use of look-up tables allows for accurate and numerically efficient run-time integration, both of which are non-trivial advantages.

**A: We have added the following sentence to the discussion to illuminate better the advantages and disadvantages of look-up tables at line 178.**

**"Lookup tables can accurately store precomputed process rates and might be numerically more efficient than analytical solutions, depending on the computer architecture, size of the lookup table, and complexity of the analytical solution."**

---

## Author Comment (AC2)

**Reviewer #2:**

This paper is concerned with the representation of ice particle aggregation in a bulk microphysics scheme. The paper uses observations from multifrequency and doppler radar to revise the necessary parameters. The methods seem to be very detailed and well-justified. Perhaps the weakest point is that the model used is 1-D and tuned to the observations, but these shortcomings are recognised and a reasonably well-described method for tuning the 1-D model is described.

Overall, I feel that the paper is an excellent contribution to the literature. It is a very dense paper, and I felt that sometimes the main message was lost a little in the text. Perhaps some of the technical detail (e.g. line 609 about the statistical metric used) could have been moved to the appendix and the main findings stated in the abstract? – at the moment this is more general findings that are stated.

I have no strong arguments against the paper, and recommend publication after considering some minor points.

**General comment by the authors to Reviewer #2:**

We thank the reviewer for the time and effort in reading the manuscript and providing constructive comments and suggestions.

We reformulated and extended the abstract to more clearly state the main findings of aggregation parameters and the proposed methodology.

In our opinion, it might be more convenient for the reader to find short definitions (e.g., of the Hellinger distance) directly in the main text without the need to read the appendix. However, we also agree that most readers might prefer a more concise text. Out of this consideration, we included only the lengthy deviations of the bulk collision rates and thermodynamic fields in the appendix and left other points in the main text.

In the following, we address the comments point by point. Line numbers refer to the non-revised manuscript.

**Point-by-point reply of the authors to the questions and comments raised by reviewer#2**

Line 33: I thought the sentence was confusing to the uninformed reader about the fallspeed of ice. This could be easily reworded to make it less ambiguous.

A: We have rephrased the sentence to describe the typical velocity-size relationship from small to large particles, which is hopefully more intuitive and understandable in this version.

"For smaller particles, vt increases strongly, but the increase in vt flattens with size and, finally, vt approaches a constant value of 1 m/s for centimeter-sized aggregates (Lohmann et al., 2016)."

Line 56: Atlas-type vt-size relation? Wasn't clear what this meant at first, but I see this is defined on page 13, line 322. Maybe this could be stated sooner.

A: We avoided the specific term "Atlas-type vt-size relations" in the revised manuscript until the point when it is properly introduced and explained. On the previous occasions, we replaced it with "more complex vt-size relations". This seems more suitable for the introduction because also other vt-size relations are imaginable that consider the asymptotic behavior of vt at large sizes. Furthermore, in the methods section, a reference to the equation of the Atlas-type vt-size relation is added.

Section 2.3: should rho\_air be included? Mass mixing ratio is kg/kg. Assuming that f(m) is number per kg per mass interval, this would mean the units of Q are kg/m-3, which is ice water content.

A: There have indeed been two mistakes in our original sentence. First, the mass density and not the mixing ratio is predicted by the scheme, which is a moment of the mass distribution (see Section 2.1 in Seifert & Beheng (2006)). Second, the mixing ratio equals the mass density divided by the air density. f(m) has units of number per kg and volume (#/(kg m^3)).

Section 2.4, line 205: DWR is calibrated to remove attenuation using disdrometer measurements. Here the authors argue that DWR is correlated to the mean mass of the distribution. I did not understand the arguments why this is the case. Could this be made clearer?

A: We revised this section strongly to illustrate the effect of mean size on DWR more clearly and clarify why we can disregard differential attenuation effects in this analysis:

"Although also differential attenuation contributes to DWR (Battaglia et al., 2020), we did not include this effect in Eq.(3) because the processing of D18 already corrects for the impact of differential attenuation on DWR. D18 evaluated the absolute calibration of the observed Ze's from the Ka-Band radar using disdrometer measurements during rainfall. After correcting differential attenuation due to gases at all three frequencies, the Ka-Band radar was then used as a reference for estimating calibration biases and differential attenuation effects due to hydrometeors by comparing the three Ze's at cloud top. The DWRs caused by differential scattering are usually close to 0 dB for small ice particles present at the cloud top. Calibration biases can be identified as DWR bias which is relatively constant over time; differential attenuation effects due to supercooled liquid water, rainfall, or the melting layer vary more strongly on shorter time scales (minutes to hours). The path integrated differential attenuation estimated at cloud top was then used to correct the DWRs in the entire profile. A more in-depth discussion of various correction methods for multi-frequency radar observations is also provided in Tridon et al. (2020). If differential scattering effects are the only contributor to DWR, it correlates well with the mean mass of the distribution f(m) (Sect. 3.1.1), as can be seen from Eq. (3). For small particles, the Rayleigh approximation is valid for all frequencies and sigma b scales with the mass squared. However, for larger particles and shorter wavelengths, sigma\_b is smaller than predicted by the Rayleigh approximation and  $\sigma_{\rm b}({\rm m},\lambda_2)$  is smaller than  $\sigma_{\rm b}({\rm m},\lambda_4)$ . As a result, particle populations that contain larger particles, e.g., due to their large mean mass, have larger DWR's than particle populations with smaller mean masses."